# Walnut (*J. regia*) Agro-Residues as a Rich Source of Phenolic Compounds

**DOI:** 10.3390/biology10060535

**Published:** 2021-06-15

**Authors:** Aljaz Medic, Jerneja Jakopic, Anita Solar, Metka Hudina, Robert Veberic

**Affiliations:** Department of Agronomy, Biotechnical Faculty, University of Ljubljana Jamnikarjeva 101, SI-1000 Ljubljana, Slovenia; jerneja.jakopic@bf.uni-lj.si (J.J.); anita.solar@bf.uni-lj.si (A.S.); metka.hudina@bf.uni-lj.si (M.H.); robert.veberic@bf.uni-lj.si (R.V.)

**Keywords:** hydrojuglone, juglone, naphthoquinones, phenolic compounds, 1,4–naphthoquinone, bark, buds, husk

## Abstract

**Simple Summary:**

Agro-residues are usually discarded as landfill, or burnt or left to decompose in the orchard. The efficient use of these walnut agro-residues would be a strategy that simultaneously helps to preserve the environment and boosts the economic outcome for farmers and companies. While some studies have reported on the content of bioactive compounds in walnut husks, little or nothing is known to date about the bioactive compounds in the buds and bark. Potentially, if walnut parts are used as a valuable source of bioactive compound, they might still be reused for other purposes. The identification and quantification of new phenolics between the different parts of the plant was carried out. It provided valuable data on their phenolic contents, and demonstrated where the extraction of individual phenolics would be meaningful. These data also show origin-related phenolic contents across the cultivars, and thus these phenolic profiles might serve to define the origins of different walnut cultivars. The study will help to propose new directions for further studies essential for agro-food, cosmetics and pharmacy industries.

**Abstract:**

The present study was designed to identify and quantify the major phenolic compounds (phenolics) in the inner and outer husks, buds and bark of the Persian walnut, *Juglans regia* L. A comparison across six different cultivars grown in Slovenia was also carried out: ‘Fernor’, ‘Fernette’, ‘Franquette’, ‘Sava’, ‘Krka’ and ‘Rubina’. A total of 83 compounds were identified, which included 25 naphthoquinones, 15 hydroxycinnamic acids, 8 hydroxybenzoic acids, 13 flavanols, 2 flavones, 1 flavanone and 19 flavonols. For the first time, 38 phenolics in the husks, 57 phenolics in the buds and 29 phenolics in the bark were presented in *J. regia* within this study. Naphthoquinones were the major phenolics determined, approximately 75% of all analysed phenolics in the inner husk, 85% in the outer husk, 50% in buds and 80% in bark. The highest content of phenolics was found in the walnut buds, followed by the bark, the inner husk and the outer husk. On the basis of these high phenolic contents, walnut husks, buds and bark represented valuable by-products of the walnut tree. These data also show origin-related phenolic contents across the cultivars, and thus these phenolic profiles might serve to define the origins of different walnut cultivars.

## 1. Introduction

Persian or English walnut (*Juglans regia* L.) is the second most cultivated tree nut worldwide. Walnuts are native to the region stretching from the Balkans eastward to the Himalayas and southwest China. Nowadays, it is widely cultivated across Europe [1]. Walnuts are the third most consumed nut in the world, after almonds (*Prunus amygdalus* Batsch) and hazelnuts (*Corylus avellana* L.) [2]. The walnut kernel represents 50% of the total fruit weight, and it is the only edible part; therefore, it is clear that a lot of walnut agro-residues are generated every year that might have the potential for further, alternative use [3]. Walnut agro-residues include: (i) the hard woody shell that protects the seed; (ii) the green husk that is rich in phenolic compounds (phenolics) and that further protects the woody shell in early stages of development; and (iii) the twigs and branches that are usually mulched after the winter pruning, with the bark and buds potentially also containing high levels of phenolics. All of these walnut agro-residues are usually discarded as landfill, or burnt or left to decompose in the orchard [4]. The efficient use of these walnut agro-residues would be a strategy that simultaneously helps to preserve the environment and boosts the economic outcome for farmers and companies [3].

As many different natural agro-residues are inexpensive and available in large quantities, there have been increasing tendencies over the last few decades towards their reuse as natural ingredients instead of chemical treatments [5]. In more recent years, several new research ideas for using walnut agro-residues have been proposed, with most studies focusing on the woody shells as a sorbent for oil [6], for hazardous material removal [7], as an ingredient in the cosmetics industry and as a blasting medium, among others [3].

The situation is quite different for walnut husks, and in particular the branches that include the buds and bark. While some studies have reported on the content of bioactive compounds in walnut husks, which have demonstrated antiradical and antimicrobial effects [3], little or nothing is known to date about the bioactive compounds in the buds and bark. Potentially, if walnut husks are used as a valuable source of bioactive compounds (i.e., mostly phenolics), they might still be reused for the removal of heavy metals from contaminated wastewaters [3,8], as a biofuel [9], for cosmetics [10] and as a natural dye [10,11]. Similarly, walnut bark might be reused as a natural dye [12], to truly make the most of these agro-residues.

Bioactive compounds are extra nutritional constituents that naturally occur in plant and food products. Most of these are secondary metabolites, such as alkaloids, pigments, mycotoxins, plant growth factors and phenolics. In recent years, numerous studies have been carried out that have promoted the benefits of such bioactive compounds for human health, in terms of potential protection against some degenerative diseases, like cancers and diabetes, and against cardiovascular diseases, and as anti-allergens, anti-microbials, anti-inflammatories and antioxidants, among others [13]. Phenolics have also been used effectively as functional ingredients in foods, as they can prevent mould and bacterial growth, and lipid oxidation [14].

Due to these countless benefits of such bioactive compounds, and of the phenolics in particular, studies have intensified to identify vegetables, fruit, plants and agricultural and agro-industrial residues as sources of phenolics. For walnuts, naphthoquinones and flavonoids have been reported to be the major phenolics [15].

Naphthoquinones are secondary metabolites that have been identified in about 20 plant families, they are most commonly found in Bignoniaceae, Droseraceae, Plumbagi-nace, Boraginaceae and Juglandaceae families, and they comprise a wide variety of chemical structures based on the naphthalene skeleton. They participate in multiple oxidative processes, serve as important links in electron transport chains, and might also act as defensive compounds in interspecies chemical warfare (i.e., alelopathy). On the basis of these traits, many studies have been exploring the biological and toxicological activities of naphthoquinones, to potentially discover and develop new drugs (e.g., antibacterial, antifungal, antiviral, antiparasitic and antitumor) [16].

Over time, microorganisms tend to develop resistance to antimicrobial agents that are used as therapeutics, which has prompted the search for new effective antimicrobials. There are numerous studies that have documented activities of a variety of naphthoquinones against an array of microorganisms, including viruses, bacteria, fungi and parasites [16]. The most studied naphthoquinones are vitamin K (e.g., anti-inflammatory, decrease of gap-junctional intercellular communication), juglone (e.g., apoptotic, cell-cycle arrest, anti-inflammatory) and plumbagin (e.g., apoptotic, cell-cycle arrest, inhibition of cell invasion, migration and proliferation, anti-inflammatory and neuroprotection) [17].

Therefore, the recovery of these secondary metabolites from walnut agro-residues might generate functional ingredients, and might also add more value to the walnut industry. To effectively recover and use the phenolics from walnut husks, buds and bark, the chemical profiles of each of these agro-residues need to be defined, especially in terms of their individual phenolics. Due to their non-specific mechanisms of action, naphthoquinones also show significant toxicity [16], which can be seen for juglone and its allelopathic effects [18]. However, adequate modifications to naphthoquinone structures might instead produce new and valuable drugs [16].

Therefore, the objective of this study was to define the phytochemical compositions of walnut husks, buds and bark, and expand the discussion on the use of bioactive molecules in these walnut agro-residues. The identification of new phenolics such as naphthoquinones, in particular, and their quantification between the different parts of the plant will provide valuable data on their phenolic contents, and it will demonstrate where the extraction of individual phenolics would be meaningful. Thus, the identification and quantification of the phenolics in different parts of walnut agro-residues might help to propose new directions for further studies essential for agro-food, cosmetics and pharmacy industries. This study follows and upgrades an earlier study by Medic et al. [19] on walnut (peeled kernel and pellicle) on the phenolic and dicarboxylic acid content of the same six cultivars.

## 2. Materials and Methods

### 2.1. Plant Materials

Samples of walnut husks, buds and bark were obtained for six walnut cultivars three French cultivars: ‘Fernor’, ‘Fernette’, ‘Franquette’ and three Slovenian cultivars: ‘Sava’, ‘Krka’ and ‘Rubina’. All of these samples were collected on 23 September 2019, at the Experimental Field for Nut Crops in Maribor (Slovenia; 46°34′01″ N; 15°37′51″ E; 275 m a.s.l.). They were obtained from 24-year-old trees at a planting density of 10 m × 10 m, with all under the same agronomical management and soil and climate conditions. The samples were collected from four trees for each cultivar, for a total of four repetitions per analysis. The samples were collected from the middle third of the branches on the east side of the trees, put in plastic bags and frozen immediately. The inner and outer husks were separated using a peeler, where ~1 mm of the husk was peeled away as outer husk, with the remaining peeled husk as the inner husk. The terms inner and outer husk are used here because the exocarp and mesocarp of the walnut husk cannot be separated completely. Therefore the inner husk represents only the husk mesocarp, while the outer husk represents the exocarp and part of the mesocarp. The samples were then transported to the laboratory of the Department of Agronomy in the Biotechnical Faculty, of the University of Ljubljana (Slovenia), where they were lyophilised, ground into a powder with liquid nitrogen, and stored at −20 °C prior to further analysis.

### 2.2. Extraction of the Individual Phenolic Compounds

Briefly, 0.25 g of inner and outer husk and bark, or 0.1 g of buds, were extracted using 100% methanol (Sigma-Aldrich, Steinheim, Germany) at a 1:20 (*w*/*v*) tissue:methanol ratio. The protocol followed that described by Medic et al. [19].

### 2.3. HPLC–Mass Spectrometry Analysis of Individual Phenolic Compounds

The phenolics were analysed on an UHPLC system (Surveyor Dionex UltiMate 3000 series; Thermo Finnigan, San Jose, CA, USA) with a diode array detector at 280 nm for hydroxycinnamic acids, hydroxybenzoic acids, flavanols, flavanones and naphthoquinones, and at 350 nm for flavones and flavonols. The spectra were recorded between 200 nm and 600 nm. A C18 column (Gemini 150 × 4.60 mm; 3 μm; Phenomenex, Torrance, CA, USA) was used to separate the phenolics, at 25 °C, as previously described by Medic et al. [19].

The identification of the phenolics was done by tandem mass spectrometry (LCQ Deca XP Max; Thermo Scientific, Waltham, MA, USA) with heated electrospray ionisation operating in negative ion mode, using the parameters as described by Medic et al. [19]. The mass spectrometry (MS) scanning for analysis was from *m/z* 50 to 2000, with data acquisition using the Xcalibur 2.2 software (Thermo Fischer Scientific Institute, Waltham, MA, USA). The phenolics were fragmented, with external standards used for the identification and quantification of known compounds. Literature data and MS fragmentation were used for identification of the unknown compounds, which were quantified using similar standards. The levels of the individual phenolics are expressed as mg/100 g dry weight, with their quantification according to the most relevant standard.

### 2.4. Analysis of Total Phenolics Content

For the full comparisons across the different walnut cultivars, the total phenolics content is represented first as the sum of all of the individual identified phenolics, each of which is expressed in mg/100 g dry weight according to the most relevant standard. A second determination of the total phenolics content was also carried out for the different walnut samples from the different walnut cultivars, with the extractions according to the same protocol as for the individual phenolics. These values for the total phenolics content of the extracts were determined using the Folin–Ciocalteau phenol reagent, as described by Singleton et al. [20], and then processed as described by Medic et al. [19] and Zamljen et al. [21]. These values are expressed in mg gallic acid equivalents/100 g dry weight.

### 2.5. Chemicals

The following standards were used to identify and quantify the phenolics: apigenin 7-glucoside, kaempferol-3-glucoside, procyanidin B1, quercetin-3-glucoside, ferulic acid, *p*-coumaric acid (Fluka Chemie GmbH, Buchs, Switzerland); (+) catechin (Roth, Karlsruhe, Germany); 4-*O*-caffeoylquinic acid, neochlorogenic acid (3-caffeoylquinic acid), myricetin-3-galactoside, quercetin-3-galactoside, quercetin-3-rhamnoside, juglone (5-hydroxy-1,4-naphthoquinone), 1,4-naphthoquinone, caffeic acid, galic acid, ellagic acid, naringenin, (−)epicatechin (Sigma–Aldrich Chemie GmbH, Steinheim, Germany); and myricetin-3-rhamnoside, quercetin-3-arabinofuranoside, quercetin-3-arabinopyranoside (Apin Chemicals, Abingdon, UK).

The water used for all sample preparation, solutions and analyses was bi-distilled and purified using a Milli-Q water purification system (Millipore, Bedford, MA, USA). The acetonitrile and formic acid for the mobile phases were HPLC-MS grade (Fluka Chemie GmbH, Buchs, Switzerland).

### 2.6. Statistical Analysis

The data were collated using Microsoft Excel 2016, and analysed using R commander. Samples of the inner and outer husks, buds and bark were assayed as four repetitions. The data are expressed as means ± standard error (SE). For determination of significant differences between the data, one-way analysis of variance (ANOVA) was used, with Tukey’s tests. Statistical means at 95% confidence level were calculated to determine the significance of the differences.

## 3. Results and Discussion

### 3.1. Identification of Individual Phenolics in Walnut Inner and Outer Husks, Buds and Bark

Based on the existing literature and the use of standard compounds, a total of 83 phenolics were tentatively identified for the inner and outer husks, buds and bark of these walnuts. Of these 83 phenolics, 14 were identified using standards, with fragmentation of both the standards and the addition of external standards used to confirm their identities. The remaining 69 phenolics were tentatively identified according to their pseudomolecular ions ([M − H]^−^) and specific fragmentation patterns (i.e., MS^2^, MS^3^, MS^4^, MS^5^). The selected MS spectra of the compounds can also be found in the Appendix A.

Most of the phenolics were identified for the buds, followed by the inner husk, the outer husk and the bark. The majority of naphthoquinones and hydroxycinnamic acids were in the inner and outer husks, and the majority of hydroxybenzoic acids and flavanols were in the buds. The only flavanone identified was in the buds.

Seven of the naphthoquinones were identified in all of the plant tissues of all of the cultivars: juglone, hydrojuglone, hydrojuglone β-D-glucopyranoside, hydrojuglone rutinoside, hydrojuglone derivative pentoside 2, hydrojuglone derivative rhamnoside and dihydroxytetralone hexoside. As well as these, the inner and outer husks contained a few hydrojuglone derivatives and mostly other naphthoquinones, while the buds and bark contained mainly hydrojuglone derivatives. To the best of our knowledge, 13 of the phenolics indicated here have not been reported for the walnut *J. regia*, or for any other *Juglans* species, or indeed, for any plant tissues: hydrojuglon, hydrojuglon rutinoside, hydrojuglone dihexoside, hydrojuglone derivative 1, hydrojuglone derivative 2, hydrojuglone derivative 3, hydrojuglone derivative 4, hydrojuglone derivative 5, hydrojuglone derivative pentoside 1, hydrojuglone derivative pentoside 2, hydrojuglone derivative pentoside 3, hydrojuglone derivative rhamnoside, hydrojuglone pentose galloyl derivative and hydrojuglone hexoside derivate.

Overall, for the inner and outer husks, 38 phenolics were identified, as 17 naphthoquinones, 11 hydroxycinnamic acids, 3 hydroxybenzoic acids, 3 flavanols, 2 flavones and 2 flavonols. For the buds, 57 phenolics were identified, as 13 naphthoquinones, 6 hydroxycinnamic acids, 6 hydroxybenzoic acids, 12 flavanols, 2 flavones, 1 flavanone and 17 flavonols. For the bark, 29 phenolics were identified, as 11 naphthoquinones, 3 hydroxybenzoic acids, 3 flavanols, 2 flavones and 10 flavonols.

Wherever possible, comparisons with authentic standards were performed. The data for all of these phenolics identified for these walnuts are summarised in Table 1 for the inner and outer husks, in Table 2 for the buds and in Table 3 for the bark.

These include mass spectrometry analysis (*m/z*, MS/MS fragmentation) and the standards according to which they were quantified.

In relation to these walnut naphthoquinones, dihydroxytetralone hexoside was identified by fragmentation ion *m/z* 159 ([M − H]^−^–H_2_O–180), and trihydroxytetralone galloyl hexoside by fragmentation ions *m/z* 331 and 271, as reported previously for walnut leaves [22]. Also, 1,4-Naphthoquinone was identified with the help of the standard at *m/z* 173, which yielded MS^2^ fragments at *m/z* 111, 155, 129 and 145, which were previously mistakenly reported as juglone in *Juglans mandshurica* [23]. Juglone was identified with the help of the standard at m/z 189, which yielded an MS^2^ fragment of *m/z* 161 and MS^3^ fragments of *m/z* 117 and 133. Hydrojuglone β-D-glucopyranoside was identified from its fragmentation that yielded an ion at *m/z* 175, defining the loss of a hexosyl moiety (-162) [24]. The MS^3^ *m/z* fragment of hydrojuglone β-D-glucopyranoside corresponded to the predicted LC-MS spectrum in a negative scan from the Human Metabolome Database, which yielded fragment ions of *m/z* 131, 157, 103 and 115.

Other phenolics identified through their fragmentation patterns included: hydrojuglone and its derivatives through the distinct fragment ions MS^n^ *m/z* 175 and MS^n+1^ *m/z* 131, 157, 103, 147 and 115, as seen for the fragmentation of hydrojuglone β-D-glucopyranoside; 5-hydroxy-2,3-dihydro-1,4-naphthalenedione through its fragmentation pattern of MS^2^ ions *m/z* 131 [M–H–CO_2_]^−^, 147 [M–H–CO]^−^, 157 [M–H–H_2_O]^−^ and 129 [M–H–CO–H_2_O]^−^; regiolone through its fragmentation pattern of MS^2^ ions *m/z* 159 [M–H–H_2_O]^−^, 175 and 131 [M–H–H_2_O–CO]^−^; 4,5,8-trihydroxynaphthalene-5-D-glucopyranoside through its fragmentation pattern of MS^2^ ions *m/z* 331 [M–H–C_10_H_8_O_3_]^−^ and 271 [M–H–C_12_H_12_O_5_]^−^, and MS^4^ ions *m/z* 211 [M–H–C_14_H_16_O_7_]^−^ and 169 [M–H–C_16_H_18_O_8_]^−^; 1,4,8-trihydroxynaphthalene-1-D-glucopyranoside through its fragmentation pattern of MS^2^ ion *m/z* 327 [M–H–C_10_H_8_O_3_]^−^ and MS^3^ ions *m/z* 183 [M–H–C_16_H_16_O_7_]^−^ and 225 [M–H–_C14H14O6_]^−^; *bis*-juglone through its fragmentation pattern of MS^2^ ions *m/z* 345 [M–H–H_2_O]^−^, 317 [M–H–H_2_O–CO]^−^ and 301 [M–H–CO_2_]^−^; and *p*-hydroxymethoxybenzobijuglone through its fragmentation pattern of MS^2^ ions *m/z* 383 [M–H–CH_4_O]^−^ and 355 [M–H–CH_4_O–CO]^−^, as reported by Huo et al. [23] in *Juglans mandshurica.* These compounds were previously reported in *J. mandshurica*, but are reported here for the first time in the walnut *J. regia*.

The 15 hydroxycinnamic acids identified through their fragmentation patterns included: neochlorogenic acid (3-caffeoylquinic acid) through its fragmentation, in addition to an external standard; 3-*p*-cumaroylquinic acid through its fragmentation pattern of MS *m/z* 337, MS^2^ *m/z* 163, 191 and 173, as reported by Liu et al. [25] and Vieira et al. [22]; *p*-coumaric acid derivatives through the *p*-coumaric acid fragmentation pattern after being broken down, through the fragmentation patterns of ions *m/z* 163 and 119, as reported by Liu et al. [25] and Vieira et al. [22]; ferulic acid derivatives through their fragmentation patterns of MS^n^ ion *m*/*z* 193 and MS^n+1^ ions *m/z* 149 and 117, as reported by Vieira et al. [22] and Šuković et al. [26]; and caffeic acid derivatives through their fragmentation pattern of MS^n^ ion *m*/*z* 179 (caffeic acid–H), as reported by Vieira et al. [22].

Seven phenolics were identified for hydroxybenzoic acids: gallic acid derivatives, through the gallic acid fragmentation pattern after being broken down, through the fragmentation pattern of ions *m/z* 169 and 125, as reported by Li and Seeram [27] and Šuković et al. [26]; *bis*-(hexahydroxydiphenoyl)-glucose through its fragmentation pattern of MS^2^ ions *m/z* 301 and 275, and MS^3^ ions *m/z* 257, 229 and 185, as reported by Medic et al. [19] and Regueiro et al. [28]; and ellagic acid derivates through the typical fragmentation ions of ellagic acid at *m*/*z* 257, 229 and 185, as reported by Singh et al. [29].

There were 13 flavanols identified through their fragmentation patterns: (+)catechin and (−)epicatechin through their fragmentation patterns, in addition to an external standard, which produced fragment ions *m/z* 245, 205 and 179 for both (+)catechin and (−)epicatechin, thus suggesting that standards are needed when determining either of those compounds; epicatechin and catechin derivatives through the (+)catechin and (−)epicatechin fragmentation patterns after being broken down, through the ions *m/z* 245, 205 and 179, as seen in standard fragmentation patterns; and procyanidin dimers and procyanidin dimer derivatives through their characteristic fragmentation of MS^n^ *m/z* 577 and MS^n+1^ *m/z* 425, 407 and 289 [14,30].

The two flavones identified were santin and 5,7-dihydroxy-3,4-dimetoxyflavone, through their fragmentation patterns according to Yan et al. [30]. Both santin and 5,7-dihydroxy-3,4-dimetoxyflavone have been reported for walnut flowers [30], and now for the first time here for walnut inner and outer husks, buds and bark.

The flavanones included the identification of one compound: naringenin, through its fragmentation in addition to an external standard, through the fragment ions *m/z* 151 and 177.

The flavonols included the identification of three groups of compounds: (i) myricetin glycosides through their fragmentation pattern of MS^2^ ions *m/z* 316, 317 and MS^3^ ions *m/z* 179, 191; (ii) quercetin and quercetin glycosides through their clear fragmentation pattern of MS^2^ m/z 301 and MS^3^ *m/z* 179, 151; and (iii) kaempferol and kaempferol glycosides through their fragmentation pattern of MS^2^ *m/z* 284 and 285 and MS^3^ *m/z* 255 and 227, as reported by Santos et al. [31] and Vieira et al. [22]. Fragmentation patterns with the loss of hexosyl (-162), pentosyl (-132) and rhamnosyl (-146) residues were seen here, as reported by Vieira et al. [22]. Kaempferol-7-hexosides were identified through their fragmentation pattern of MS^2^ ion *m/z* 285 and MS^3^ ions 165, 119 and 93, as reported by Chen et al. [32]. Kaempferol-7-hexosides have been reported previously for *Rhamnus davurica* [32], but this is the first time for walnut.

### 3.2. Quantification of Total and Individual Phenolic Compounds for Walnut Inner and Outer Husks, Buds and Bark

The highest contents of phenolics were in the walnut buds, followed by the bark, the inner husk and the outer husk, as shown in Figure 1B.

The highest relative contents of hydroxycinnamic acids and flavones were seen for the inner husk, with the highest relative contents of hydroxybenzoic acids, flavanols, flavanones and flavonols for the walnut buds, as shown in Figure 1A. The higher absolute contents of phenolics in the walnut buds compared to the bark was mostly because of the higher content of flavanols, flavonols, hydroxycinnamic and hydroxybenzoic acids in the buds. The content of naphthoquinones was around 11 to 12 g/100 g plant material in both plant tissues. Therefore, walnut buds and bark represent an excellent source of naphthoquinones.

The total naphthoquinones were the major phenolic group determined for the inner and outer husks, and for the buds and bark as well. These represented approximately just over 50% of all of the identified phenolics in the buds, 75% in the inner husk, 80% in the bark and 85 % in the outer husk, as shown in Figure 1A. As mentioned above, various naphthoquinones have shown activities against an array of microorganisms, including viruses, bacteria, fungi and parasites [16]. While the walnut buds were a better source of flavanols, hydroxybenzoic acids and flavonols, the inner and outer husks can also be considered as a source of naphthoquinones, with different naphthoquinones in the walnut buds and bark compared to the inner and outer husks, as seen in Table 1.

While the content of phenolics is usually higher in the peel of fruit compared to the flesh [19], here, interestingly, the content of phenolics for the outer husk was much lower than for the inner husk. When considering further the different tissues of walnut plants, the total phenolic contents (both as the summation and the total extracts; Table 4, Table 5, Table 6 and Table 7) were higher than any previously reported for walnut shoots [15], leaves [18] or kernels [19], which further justifies the use of the husk, buds and bark as sources of the phenolics.

Interestingly, the three Slovenian cultivars of ‘Sava’, ‘Krka’ and ‘Rubina’ had similar naphthoquinone contents in the walnut outer husk that were also higher than for the French cultivars ‘Fernor’, ‘Fernette’ and ‘Franquette’, which were also similar for their total naphthoquinone contents. The same was seen for the walnut inner husk, where the Slovenian varieties showed higher total naphthoquinone content than the French cultivars. This information that the Slovenian cultivars had higher total naphthoquinone contents than the French cultivars might also be useful in the future determination of the genetic origins of a cultivar, as cultivars that are bred in different climates might have specific naphthoquinone contents, as previously reported by Medic et al. [19]. The summarised total phenolic content of the buds identified in this study was compared with the total phenolic content of pellicle identified by Medic et al. [19], which was similar in terms of the order of phenolic compound content of the selected cultivars, with ‘Franquette’ containing the most phenols, followed by ‘Fernor’, ‘Krka’ and ‘Sava’ and ‘Rubina’ containing the second least phenols and the least ‘Fernette’. Otherwise, no clear picture was seen linking phenolic content to different plant organs, suggesting that the total phenolics analysed may not be related between different parts of walnut, but rather a characteristic of the cultivar that dictates where the majority of phenolics are concentrated. Of note, this was observed only for the inner and outer husks, and not for the buds or bark, where the total naphthoquinone contents were not influenced by the origins of the cultivars.

Looking at individual naphthoquinones, in all these plant tissues, juglone was most abundant in the walnut inner husk, as can be seen in Table 4, Table 5, Table 6 and Table 7. Among the walnut cultivars, ‘Rubina’ had the highest juglone content for the inner and outer husks and the bark, and the second highest juglone content for the buds, following ‘Fernor’. This makes ‘Rubina’ an excellent choice for the purpose of juglone extraction. As juglone is used as a natural dye [12] and has anti-inflammatory effects [17], the efficient use of these agro-residues would represent a strategy that simultaneously helps to preserve the environment and potentially to boost the economic outcome for farmers and companies. This calls for further studies on the extraction of juglone from these, and other, plant tissues. The 83 phenolics identified across the different parts of the walnut tissues are shown in Table 8.

## 4. Conclusions

A total of 83 individual phenolics and the total phenolics content were identified and quantified for the inner and outer husks, buds and bark of six walnut cultivars. These 83 phenolics comprised 25 naphthoquinones, 15 hydroxycinnamic acids, 8 hydroxybenzoic acids, 13 flavanols, 2 flavones, 1 flavanone and 19 flavonols. Thirteen naphthoquinones have been reported for the walnut *J. regia*, or any other species for the first time, that may be unique to Juglandaceae family. To the best of our knowledge, this is the most complete study to describe the levels of the various phenolics for walnut husk, buds and bark. Furthermore, this is the first report to provide not only characterisation and quantification of the phenolics for walnut buds, but also a detailed characterisation and quantification of the separate husk layers (i.e., inner, outer). These data demonstrate the levels of the phenolics in these different walnut tissues, which are classified as agro-residuals. When considering the different walnut tissues, the total phenolic contents (both for the sum, and for the total extracts) were higher than previously reported for walnut shoots, leaves and kernels. This justifies the use of the husk, buds and bark as sources of phenolics. Furthermore, the Slovenian cultivars showed higher total naphthoquinone contents in the outer and inner walnut husks compared to the French cultivars. This information might be useful for the future determination of the genetic origin of a cultivar and also for the authentication of the walnuts belonging to each region, country, etc., as cultivars bred in different climates appear to show some specific variations in their naphthoquinone contents. Consequently, the present study provides useful information not only for agro-food industry (additives, pesticides) but also for the cosmetic and pharmaceutical industries. 

## Figures and Tables

**Figure 1 biology-10-00535-f001:**
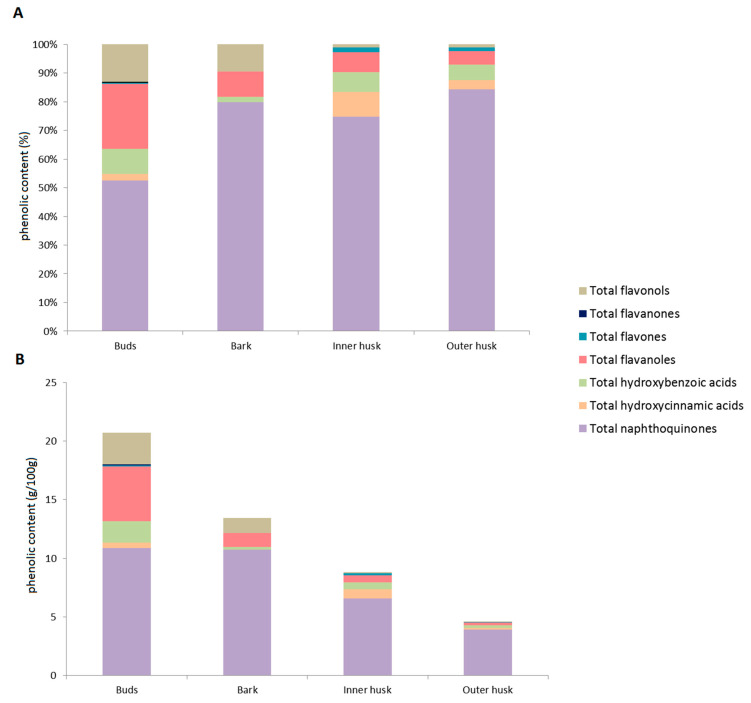
Relative contents of the phenolics groups for the different walnut tissues, as proportions of total phenolic compounds identified (**A**) and as g/100 g walnut tissue defined by the most relevant standards (**B**).

**Table 1 biology-10-00535-t001:** Tentative identification of the 38 phenolics from the walnut inner and outer husks, and the standard equivalents used.

Phenolic	Rt	[M − H]^−^	Fragmentation Pattern (*m/z*)				Equivalents Expressed
	(min)	(*m/z*)	MS^2^	MS^3^	MS^4^	MS^5^	
*p*-Coumaric acid derivative 2	7.06	343	163, 325, 119				*p*-Coumaric acid
Ferulic acid derivative 1	8.20	221	149, 117				Ferulic acid
Neochlorogenic acid (3-caffeoylquinic acid)	9.90	353	191, 179, 135				Neochlorogenic acid
Ferulic acid derivative 2	11.43	489	193	149, 134, 178, 117			Ferulic acid
(+)Catechin	12.34	289	245, 205, 179, 125				(+)Catechin
3-*p*-Coumaroylquinic acid	12.34	337	163, 191, 173				4-*O*-Caffeoylquinic acid
Dihydroxytetralone hexoside	12.69	339	159, 177	115, 75			Juglone
Caffeic acid derivative 2	13.95	251	207, 175	179, 191			Caffeic acid
5-Hydroxy-2,3-dihydro-1,4-naphthalenedione	13.97	175	131, 147, 157, 115, 103, 129				Juglone
*p*-Coumaric acid derivative 3	14.51	325	235, 265, 163	163, 191, 119			*p*-Coumaric acid
Regiolone	14.79	177	**159**, 175, 131	115, 75			Juglone
			159, **175**, 131	131, 147, 157, 103, 129, 119			
(−)Epicatechin	14.79	289	245, 205, 179, 125				(−)Epicatechin
Gallic acid derivative 4	15.87	261	243, 201, 187	199, 225, 169, 125			Gallic acid
Hydrojuglone β-D-glucopyranoside	16.49	337	175	131, 147, 157, 103			Juglone
Trihydroxytetralone galloyl hexoside	17.77	507	331, 271	271			Juglone
Hydrojuglone derivative rhamnoside	18.02	449	303, 285	285	**241**, 175, 257	213, 199, 197	Juglone
					241, **175**, 257	157, 147, 129, 119	
Quercetin-3-galactoside	18.23	463	301	179, 151			Quercetin-3-galactoside
*p*-Coumaric acid derivative 4	19.64	475	265, 163	205, 163, 119			*p*-Coumaric acid
Gallic acid derivative 5	19.80	421	313, 169	169, 125			Gallic acid
(epi)Catechin derivative 5	24.04	469	289	245, 205, 179, 125			(+)Catechin
4,5,8-Trihydroxynaphthalene-5-D-glucopyranoside	20.55	507	331, 271	271	211, 169		Juglone
Ferulic acid derivative 3	21.03	521	473, 503, 337	337	193, 175		Ferulic acid
Hydrojuglone derivative pentoside 2	21.42	435	303, 285	285			Juglone
Caffeic acid derivative 3	21.78	519	489	161, 313, 179			Caffeic acid
Gallic acid derivative 3	22.10	489	271, 313	211, 169, 125			Gallic acid
Quercetin-3-rhamnoside	22.60	447	301	179, 151			Quercetin-3-rhamnoside
1,4,8-Trihydroxynaphthalene-1-D-glucopyranoside	24.11	503	327	183, 225, 139			Juglone
Hydrojuglone hexoside derivative	24.86	497	335	175	131, 157, 103, 147		Juglone
Hydrojuglone derivative 5	26.29	517	175, 341	131, 157, 103, 147			Juglone
Caffeic acid derivative 4	26.62	499	341, 323, 281, 175	251, 281, 179	179		Caffeic acid
Hydrojuglone	28.57	175	131, 103, 157, 175				Juglone
1,4-Naphthoquinone	28.57	173	111, 155, 129, 145				1,4-Naphthoquinone
Hydrojuglone rutinoside	29.56	483	175, 325	131, 157, 103, 147			Juglone
Juglone	30.05	189	161	117, 133			Juglone
*bis*-Juglone	31.42	363	345, 317, 319, 301	301			Juglone
*p*-Hydroxymetoxybenzobijuglone	31.70	415	383, 355				Juglone
Santin	32.37	343	328	313, 285			Apigenin-7-glucoside
5,7-Dihydroxy-3,4-dimetoxyflavone	32.60	313	298	283, 255			Apigenin-7-glucoside

Rt, retention time; [M − H]^–^, pseudo-molecular ion identified in negative ion mode; **bold** numbers, fragments further fragmented; first fragment number, fragment that was further fragmented if no bold numbers are given.

**Table 2 biology-10-00535-t002:** Tentative identification of the 57 phenolics from the walnut buds, and the standard equivalents used.

Phenolic	Rt	[M − H]^−^	Fragmentation Pattern (*m/z*)				Equivalents Expressed
	(min)	(*m/z*)	MS^2^	MS^3^	MS^4^	MS^5^	
Gallic acid derivative 1	8.01	345	169, 125, 175	125			Gallic acid
Procyanidin dimer derivative 1	8.84	593	425, 467, 407, 289				Procyanidin B1
Neochlorogenic acid (3-caffeoylquinic acid)	9.75	353	191, 179, 135				Neochlorogenic acid
(epi)Catechin derivative 1	9.90	357	289, 311	245, 205, 179, 125			(+)Catechin
*bis*-HHDP-glucose	10.40	783	301, 275	257, 229, 185			Gallic acid
Procyanidin dimer 1	10.58	577	425, 407, 451, 289				Procyanidin B1
Procyanidin dimer 2	11.62	577	425, 407, 451, 289				Procyanidin B1
(+)Catechin	12.45	289	245, 205, 179, 125				(+)Catechin
3-*p*-Coumaroylquinic acid	12.55	337	163, 191, 173				4-*O*-Caffeoylquinic acid
Dihydroxytetralone hexoside	12.76	339	177, 159				Juglone
(epi)Catechin derivative 2	13.33	325	289, 163, 179	245, 205, 179, 125			(+)Catechin
Hydrojuglone dihexoside	13.89	499	175	131, 157, 103, 129, 147			Juglone
Hydrojuglone derivative 1	14.44	355	175, 169, 265, 193	131, 147, 157, 103, 129			Juglone
(−)Epicatechin	14.74	289	245, 205, 179, 125				(−)Epicatechin
(epi)Catechin derivative 3	14.74	325	163, 179, 289, 119	245, 205, 179, 125			(+)Catechin
*p*-Coumaric acid derivative	15.33	281	163, 135, 119	119			*p*-Coumaric acid
Hydrojuglone β-D-glucopyranoside	16.54	337	175	131, 103, 157, 145			Juglone
Procyanidin dimer derivative 2	17.15	729	577	425, 407, 451, 289			Procyanidin B1
Ellagic acid derivative	17.42	467	391, 301	301	257, 229, 185		Ellagic acid
Hydrojuglone derivative 2	17.98	465	**301**, 339	**215**, 257, 283, 175, 151	187, 171, 143		Juglone
				215, 257, 283, **175**, 151	147, 131, 157, 129		
Myricetin-3-galactoside	18.13	479	316	271, 287, 179			Myricetin-3-galactoside
Hydrojuglone derivative pentoside 1	18.51	435	285	241, 199, 175, 151, 135	213, 199, 197		Juglone
Gallic acid derivative 2	19.28	491	271, 331	211, 169, 125			Gallic acid
Quercetin galoyll hexoside	19.71	615	463	301	179, 151		Quercetin-3-glucoside
Myricetin pentoside	19.99	449	317, 316	179, 151, 191			Myricetin-3-galactoside
Galloyl-3-(epi)catechin	20.36	441	289	245, 205, 179, 125			(+)Catechin
Quercetin-3-galactoside	20.36	463	301	179, 151			Quercetin-3-galactoside
Myricetin-3-rhamnoside	20.61	463	316	271, 287, 179, 151			Myricetin-3-rhamnoside
Quercetin-3-glucoside	20.80	463	301	179, 151			Quercetin-3-glucoside
Hydrojuglone derivative rhamnoside	21.17	449	303, 285				Juglone
Hydrojuglone derivative pentoside 2	21.48	435	303, 285	285, 177	241, 175, 161	213, 199, 197	Juglone
Gallic acid methyl ester	21.86	183	168, 124	124	95		Gallic acid
Quercetin-3-arabinopyranoside	21.99	433	301	179, 151			Quercetin-3-rabinopyranoside
Gallic acid derivative 3	22.17	489	271	211, 169	168, 124		Gallic acid
Quercetin-3-arabinofuranoside	22.41	433	301	179, 151			Quercetin-3-arabinofuranoside
Quercetin-3-rhamnoside	22.55	447	301	179, 151			Quercetin-3-rhamnoside
Kaempferol pentoside 1	23.22	417	284	255, 227, 151			Kaempferol-3-glucoside
Kaempferol pentoside 2	23.48	417	284	255, 227, 151			Kaempferol-3-glucoside
Caffeic acid hexoside derivative	23.84	501	341	251, 281, 179, 323	179, 135		Caffeic acid
Kaempferol pentoside 3	24.06	417	285	257, 267, 229			Kaempferol-3-glucoside
Kaempferol-3-rhamnoside	24.32	431	285	257, 268, 229			Kaempferol-3-glucoside
(epi)Catechin derivative 4	24.56	463	289	245, 205, 179	203, 187, 161		(+)Catechin
Hhydrojuglone pentose galloyl derivative	24.85	587	455	303, 285	285, 259, 177, 241, 175		Juglone
Quercetin hexoside derivative 1	25.13	669	463	301	179, 151		Quercetin-3-glucoside
Procyanidin dimer derivative 3	25.71	903	729	603, 577			Procyanidin B1
Quercetin hexoside derivative 2	26.15	639	463	301	179, 151		Quercetin-3-glucoside
Caffeic acid derivative 1	26.62	499	341, 323, 281, 175	251, 221, 179	179, 135		Caffeic acid
Diferuoyl hexoside	27.73	531	337	193, 178	134, 149		Ferulic acid
Hydrojuglone	28.60	175	131, 103, 157, 175				Juglone
Quercetin	29.34	301	179, 151				Quercetin-3-glucoside
Hydrojuglone rutinoside	29.63	483	175	131, 157, 103			Juglone
Hydrojuglone derivative 3	29.73	513	175, 337	131, 157, 103			Juglone
Juglone	30.14	189	161	117, 133			Juglone
Naringenin	30.33	271	151, 177				Naringenin
Kaempferol	30.64	285	151				Kaempferol-3-glucoside
Santin	31.97	343	328	313, 285			Apigenin-7-glucoside
5,7-Dihydroxy-3,4-dimetoxyflavone	32.34	313	298	283, 255			Apigenin-7-glucoside

Rt, retention time; [M − H]^–^, pseudo-molecular ion identified in negative ion mode; **bold** numbers, fragments further fragmented; first fragment number, fragments that were further fragmented if no bold numbers are given.

**Table 3 biology-10-00535-t003:** Tentative identification of the 29 phenolics from the walnut bark, and the standard equivalents used.

Phenolic	Rt	[M − H]^−^	Fragmentation Pattern (*m/z*)			Equivalents Expressed
	(min)	(*m/z*)	MS^2^	MS^3^	MS^4^	
Procyanidin dimer 2	11.67	577	425, 407, 451, 289			Procyanidin B1
(+)Catechin	12.43	289	245, 205, 179, 125			(+)Catechin
Dihydroxytetralone hexoside	12.82	339	177, 159			Juglone
Hydrojuglone β-D-glucopyranoside	16.58	337	175	131, 147, 157, 103		Juglone
Procyanidin dimer derivative 2	17.16	729	577	425, 407, 451, 289		Procyanidin B1
Ellagic acid derivative	17.47	467	391, 301	301	257, 229, 185	Ellagic acid
Hydrojuglone derivative 4	17.83	451	**319**, 325, 301, 193, 151	193, 301, 179, 125	165, 175, 121, 131	Juglone
			319, **325**, 301, 193, 151	192, 235		
			319, 325, **301**, 193, 151	215, 257, 283, **175**, 151	147, 131, 157, 129	
			319, 325, **301**, 193, 151	**215**, 257, 283, 175, 151	187, 171, 143	
Hydrojuglone derivative 2	18.02	465	301, 339, 319, 151, 193	215, 257, 283, 175, 151	187, 171, 143	Juglone
Hydrojuglone derivative pentoside 1	18.21	435	**285**, 301	241, 175, 199, 257, 151		Juglone
			285, **301**	229, 179, 151, 257, 137		
Hydrojuglone derivative pentoside 3	18.55	435	285, 301			Juglone
Myricetin pentoside	19.20	449	317	179, 151		Myricetin-3-galactoside
Gallic acid derivative 2	19.32	491	271	211, 169, 125	168, 124	Gallic acid
Quercetin galloyl hexoside	19.74	615	463	301	179, 151	Quercetin-3-glucoside
Myricetin-3-rhamnoside	20.33	463	316	271, 287, 179, 164	243, 227, 215, 183	Myricetin-3-rhamnoside
Quercetin-3-galactoside	20.64	463	301	179, 151		Quercetin-3-galactoside
Quercetin-3-glucoside	20.84	463	301	179, 151		Quercetin-3-glucoside
Hydrojuglone derivative rhamnoside	21.22	449	**303**, 285	181, 153, 285		Juglone
			303, **285**	241, 175, 257, 199, 151		
Hydrojuglone derivative pentoside 2	21.47	435	285	241, 175, 257		Juglone
Quercetin-3-arabinopyranoside	22.04	433	301	179, 151		Quercetin-3-arabinopyranoside
Gallic acid derivative 3	22.19	489	**271**, 313	211, 169, 125	168, 124	Gallic acid
			271, **313**	169, 125	125	
Quercetin-3-arabinofuranoside	22.39	433	301	179, 151		Quercetin-3-arabinofuranoside
Quercetin-3-rhamnoside	22.67	447	301	179, 151, 273, 257, 229		Quercetin-3-rhamnoside
Kaempferol-7-hexoside 1	23.06	447	285	165, 119, 93		Kaempferol-3-glucoside
Kaempferol-7-hexoside 2	28.34	447	285	165, 119, 93		Kaempferol-3-glucoside
Hydrojuglone	28.56	175	131, 103, 157, 175			Juglone
Hydrojuglone rutinoside	29.64	483	175	131, 103, 157		Juglone
Juglone	30.14	189	161	117, 133		Juglone
Santin	31.98	343	328	313, 285		Apigenin-7-glucoside
5,7-Dihydroxy-3,4-dimetoxyflavone	32.34	313	298	283, 255		Apigenin-7-glucoside

Rt, retention time; [M − H]^–^, pseudo-molecular ion identified in negative ion mode; **bold** numbers, fragments further fragmented; first fragment number, fragments that were further fragmented if no bold numbers are given.

**Table 4 biology-10-00535-t004:** Individual phenolics for the walnut inner husks across the six selected cultivars.

Phenolic	Inner Husk Phenolic Content per Cultivar (mg/100 g Dry Weight)					
	‘Fernor’	‘Fernette’	‘Franquette’	‘Sava’	‘Krka’	‘Rubina’
**Naphthoquinones**						
1,4-Naphthoquinone	1148.5 ± 109.0 b	1067.5 ± 95.2 b	435.3 ± 73.6 a	1521.8 ± 47.3 b	2231.8 ± 233.2 c	1547.1 ± 105.5 b
Juglone	593.2 ± 45.9 a	776.3 ± 39.8 ab	533.6 ± 34.3 a	598.9 ± 61.8 a	852.1 ± 93.4 b	852.0 ± 38.8 b
Hydrojuglone	143.2 ± 13.6 b	133.1 ± 11.9 b	54.3 ± 9.2 a	189.8 ± 5.9 b	278.3 ± 29.1 c	192.9 ± 13.2 b
Hydrojuglone β-D-glucopyranoside	502.9 ± 9.9 b	250.9 ± 17.4 a	266.3 ± 21.4 a	761.9 ± 50.1 c	624.9 ± 31.8 bc	734.7 ± 69.4 c
Hydrojuglone rutinoside	125.2 ± 2.3 ab	93.4 ± 10.6 a	90.7 ± 14.7 a	179.2 ± 14.7 bc	234.1 ± 17.3 c	200.8 ± 15.5 c
Hydrojuglone derivative 5	153.9 ± 7.0 b	75.3 ± 5.1 a	58.3 ± 10.3 a	294.1 ± 31.3 c	286.2 ± 12.4 c	187.2 ± 22.2 b
Hydrojuglone derivative pentoside 2	154.8 ± 16.6 bc	95.9 ± 9.6 ab	65.4 ± 11.8 a	246.7 ± 23.2 d	236.0 ± 21.7 cd	274.2 ± 22.2 d
Hydrojuglone derivative rhamnoside	324.2 ± 8.4 bc	155.7 ± 11.6 a	208.8 ± 25.2 ab	572.1 ± 45.0 e	486.3 ± 45.0 de	403.8 ± 45.1 cd
Hydrojuglone hexoside derivative	137.9 ± 13.6 ab	97.6 ± 11.8 a	85.2 ± 20.2 a	239.9 ± 16.4 c	194.5 ± 26.3 bc	153.6 ± 19.7 ab
*bis*-Juglone	110.7 ± 7.0 a	125.8 ± 11.0 a	133.8 ± 17.8 ab	247.9 ± 16.0 c	194.4 ± 19.1 bc	175.6 ± 13.2 ab
*p*-Hydroxymetoxybenzobijuglone	680.4 ± 92.1 a	494.3 ± 85.9 a	440.1 ± 46.6 a	490.5 ± 60.1 a	641.6 ± 8.8 a	528.1 ± 46.9 a
Regiolone	672.3 ± 18.6 ab	511.3 ± 39.8 a	775.1 ± 73.8 bc	808.6 ± 44.6 bc	936.3 ± 42.8 c	920.9 ± 38.8 c
4,5,8-Trihydroxynaphthalene-5-D-glucopyranoside	709.0 ± 44.7 bc	479.5 ± 32.6 ab	412.6 ± 15.3 a	1276.7 ± 45.1 d	1241.5 ± 58.5 d	913.6 ± 88.1 c
1,4,8-Trihydroxynaphthalene-1-D-glucopyranoside	335.7 ± 42.6 bc	151.1 ± 11.3 a	156.3 ± 21.3 a	478.0 ± 50.7 c	339.1 ± 23.2 bc	262.1 ± 23.6 ab
Dihydroxytetralone hexoside	88.2 ± 3.6 b	21.4 ± 9.3 a	68.9 ± 14.7 ab	106.8 ± 16.7 b	113.4 ± 11.9 b	104.2 ± 12.2 b
Trihydroxytetralone galloyl hexoside	169.6 ± 10.2 b	61.2 ± 10.4 a	78.4 ± 20.7 a	240.8 ± 27.2 b	182.7 ± 12.7 b	174.0 ± 23.1 b
**Hydroxycinnamic acids**						
Neochlorogenic acid (3-caffeoylquinic acid)	28.5 ± 0.8 ab	17.0 ± 1.2 a	21.6 ± 1.0 ab	47.6 ± 5.3 d	43.0 ± 2.6 cd	31.9 ± 3.5 bc
3-*p*-Coumaroylquinic acid	494.7 ± 4.3 a	496.9 ± 26.7 a	443.4 ± 18.4 a	712.2 ± 26.1 b	628.7 ± 32.1 b	474.2 ± 17.8 a
*p*-Coumaric acid derivative 2	11.6 ± 0.5 ab	8.8 ± 0.8 a	10.8 ± 0.8 ab	21.6 ± 1.5 d	17.7 ± 0.6 cd	14.4 ± 1.4 bc
*p*-Coumaric acid derivative 3	16.7 ± 0.6 ab	12.5 ± 1.1 a	17.8 ± 1.3 ac	20.3 ± 2.5 bc	22.0 ± 1.0 bc	23.8 ± 1.5 c
*p*-Coumaric acid derivative 4	11.1 ± 0.3 b	6.5 ± 0.5 a	4.8 ± 0.6 a	13.3 ± 0.9 b	11.8 ± 0.6 b	12.2 ± 1.2 b
Caffeic acid derivative 2	9.5 ± 0.4 ab	5.0 ± 1.4 a	9.3 ± 1.6 ab	14.1 ± 1.6 b	14.9 ± 0.9 b	13.1 ± 1.3 b
Caffeic acid derivative 3	52.9 ± 3.0 b	29.7 ± 2.3 a	25.1 ± 2.9 a	85.2 ± 4.2 c	61.4 ± 4.8 b	64.3 ± 5.6 b
Caffeic acid derivative 4	4.8 ± 0.5 ab	3.2 ± 0.8 a	2.8 ± 0.9 a	5.4 ± 0.9 ab	7.2 ± 1.3 b	4.9 ± 0.8 ab
Ferulic acid derivative 1	29.8 ± 3.1 bc	10.5 ± 1.1 a	18.3 ± 2.0 ab	34.8 ± 2.7 cd	40.3 ± 2.2 cd	43.7 ± 4.3 d
Ferulic acid derivative 2	19.8 ± 1.3 ab	12.4 ± 0.6 a	18.2 ± 1.3 a	34.5 ± 3.0 c	27.0 ± 1.2 b	17.8 ± 2.0 a
Ferulic acid derivative 3	15.2 ± 1.5 a	11.2 ± 0.4 a	10.5 ± 1.5 a	28.9 ± 1.8 b	26.9 ± 1.1 b	28.6 ± 2.2 b
**Hydroxybenzoic acids**						
Gallic acid derivative 3	335.7 ± 9.9 b	148.7 ± 10.4 a	240.6 ± 14.6 ab	897.8 ± 41.4 d	766.9 ± 27.5 c	817.6 ± 31.2 cd
Gallic acid derivative 4	53.8 ± 3.1 b	21.3 ± 2.2 a	30.7 ± 2.5 a	41.4 ± 7.4 ab	42.9 ± 5.8 ab	42.6 ± 6.4 ab
Gallic acid derivative 5	26.4 ± 1.9 bc	21.0 ± 2.0 ab	12.7 ± 2.4 a	38.8 ± 5.0 cd	44.8 ± 2.9 d	31.9 ± 1.6 bc
**Flavanols**						
(+)Catechin	151.7 ± 1.0 a	151.5 ± 8.1 a	136.1 ± 5.7 a	218.1 ± 8.1 b	192.1 ± 10.0 b	145.4 ± 5.4 a
(−)Epicatechin	398.9 ± 9.9 ab	305.1 ± 23.8 a	459.9 ± 44.9 bc	480.6 ± 27.0 bc	557.7 ± 25.7 c	546.6 ± 23.7 c
**Flavones**						
Santin	61.3 ± 6.1 ab	51.8 ± 1.3 a	56.1 ± 4.7 a	92.5 ± 11.4 c	89.5 ± 3.5 bc	51.2 ± 6.7 a
5,7-Dihydroxy-3,4-dimetoxyflavone	43.1 ± 1.7 a	30.4 ± 5.2 a	23.1 ± 5.4 a	166.8 ± 5.7 d	124.4 ± 11.7 c	74.3 ± 6.0 b
**Flavonols**						
Quercetin-3-galactoside	72.3 ± 3.1 abc	49.2 ± 6.5 ab	40.6 ± 3.5 a	115.0 ± 15.3 d	91.0 ± 10.1 cd	85.1 ± 7.6 bd
Quercetin-3-rhamnoside	18.3 ± 0.7 bc	7.7 ± 2.1 ab	6.3 ± 3.0 a	22.1 ± 3.0 c	24.5 ± 1.5 c	24.4 ± 4.2 c
**Total naphthoquinones**	6049.6 ± 224.4 b	4590.5 ± 193.6 a	3863.0 ± 294.6 a	8253.6 ± 192.3 cd	9073.1 ± 301.2 d	7624.8 ± 361.0 c
**Total hydroxycinnamic acids**	694.6 ± 5.0 ab	613.6 ± 27.4 ab	582.7 ± 22.6 a	1017.7 ± 35.0 c	900.2 ± 40.6 c	728.9 ± 39.4 b
**Total hydroxybenzoic acids**	415.8 ± 12.1 b	190.9 ± 9.7 a	284.0 ± 17.2 a	977.9 ± 42.3 c	854.6 ± 32.5 c	892.1 ± 37.8 c
**Total flavanols**	550.6 ± 9.4 ab	456.6 ± 22.7 a	596.1 ± 45.9 bc	698.7 ± 31.3 cd	749.8 ± 18.6 d	692.0 ± 19.9 cd
**Total flavones**	104.4 ± 7.7 a	82.2 ± 5.7 a	79.2 ± 9.6 a	259.3 ± 16.4 b	213.9 ± 13.8 b	125.5 ± 2.0 a
**Total flavanones**	nd	nd	nd	nd	Nd	nd
**Total flavonols**	90.6 ± 3.8 bc	56.9 ± 6.5 ab	46.9 ± 6.4 a	137.1 ± 15.2 d	115.5 ± 9.3 cd	109.5 ± 11.5 cd
**Total phenolics content (summation; relevant standards) ^x^**	7905.5 ± 236.8 b	5990.8 ± 204.2 a	5451.8 ± 369.6 a	11344.4 ± 298.0 cd	11907.1 ± 228.9 d	10172.8 ± 426.0 c
**Total phenolics content (total extracts; mg gallic acid equivalents/100 g dry weight) ^y^**	1447.2 ± 73.7 ab	1327.2 ± 92.9 a	1575.1 ± 92.0 ab	1842.6 ± 175.9 b	1804.5 ± 26.5 b	1788.1 ± 85.5 b

Data are means ± standard error. ^x^ expressed as the sum of all of the individual identified phenolics (summation), in mg/100 g dry weight of the most relevant standard. ^y^ expressed as the separate analysis of the total phenolics for each extract (total extracts), in mg gallic acid equivalents/100 g dry weight. Means followed by different letters within a cultivar are significantly different (*p* ≤ 0.05; Tukey’s tests); nd, not detected.

**Table 5 biology-10-00535-t005:** Individual phenolics for the walnut outer husks across the six selected cultivars.

Phenolic	Outer Husk Phenolic Content per Cultivar (mg/100 g Dry Weight)					
	‘Fernor’	‘Fernette’	‘Franquette’	‘Sava’	‘Krka’	‘Rubina’
**Naphthoquinones**						
1,4-Naphthoquinone	128.8 ± 20.4 a	260.6 ± 47.9 a	296.5 ± 34.3 a	800.8 ± 55.3 b	1079.9 ± 44.7 c	1026.5 ± 117.7 bc
Juglone	431.7 ± 25.1 a	528.5 ± 37.1 ab	519.8 ± 8.7 ab	609.2 ± 18.7 b	608.3 ± 9.6 b	838.8 ± 19.5 c
Hydrojuglone	25.7 ± 4.1 a	52.0 ± 9.6 a	59.2 ± 6.8 a	159.8 ± 11.0 b	215.5 ± 8.9 c	204.8 ± 23.5 bc
Hydrojuglone β-D-glucopyranoside	234.1 ± 9.8 a	218.5 ± 21.7 a	291.3 ± 24.6 ab	361.7 ± 35.2 b	278.7 ± 15.9 ab	370.0 ± 19.5 b
Hydrojuglone rutinoside	82.8 ± 7.3 ab	80.3 ± 8.9 a	70.2 ± 2.9 a	97.6 ± 11.0 ab	116.7 ± 4.3 bc	132.2 ± 8.5 c
Hydrojuglone derivative 5	70.5 ± 14.6 a	49.0 ± 3.3 a	54.8 ± 4.7 a	165.2 ± 18.8 b	178.1 ± 16.6 b	173.2 ± 6.2 b
Hydrojuglone derivative pentoside 2	545.7 ± 40.8 c	493.5 ± 54.4 bc	327.9 ± 27.3 a	480.2 ± 20.1 bc	371.9 ± 11.6 ab	328.8 ± 12.4 a
Hydrojuglone derivative rhamnoside	172.5 ± 17.8 bc	109.3 ± 11.5 a	164.4 ± 10.3 ab	202.2 ± 16.7 bd	224.5 ± 13.1 cd	244.7 ± 3.8 d
Hydrojuglone hexoside derivative	107.8 ± 6.3 a	101.8 ± 13.6 a	134.1 ± 10.5 a	79.7 ± 15.1 a	75.9 ± 16.3 a	124.2 ± 16.7 a
*bis*-Juglone	102.1 ± 9.8 a	150.6 ± 18.4 ab	132.6 ± 9.4 ab	177.5 ± 14.5 bc	150.4 ± 9.1 ab	215.3 ± 19.6 c
*p*-Hydroxymetoxybenzobijuglone	128.7 ± 16.3 a	133.0 ± 19.8 a	168.3 ± 14.7 ab	201.7 ± 15.4 ab	193.9 ± 20.4 ab	237.6 ± 20.5 b
Regiolone	162.3 ± 11.8 a	131.7 ± 11.7 a	153.8 ± 9.1 a	128.1 ± 9.2 a	245.8 ± 12.3 b	326.0 ± 22.0 c
5-Hydroxy-2,3-dihydro-1,4-naphthalenedione	79.1 ± 11.6 ab	120.1 ± 6.5 ab	126.0 ± 19.6 b	70.5 ± 9.5 a	91.5 ± 9.8 ab	103.9 ± 4.2 ab
4,5,8-Trihydroxynaphthalene-5-D-glucopyranoside	459.8 ± 15.0 ab	337.8 ± 30.3 a	418.8 ± 28.0 ab	672.3 ± 25.4 c	529.1 ± 26.3 b	392.3 ± 35.4 a
1,4,8-Trihydroxynaphthalene-1-D-glucopyranoside	87.4 ± 8.4 a	100.6 ± 14.3 ab	119.0 ± 15.2 ab	104.3 ± 19.1 ab	161.6 ± 14.7 b	148.5 ± 8.3 ab
Dihydroxytetralone hexoside	58.9 ± 9.0 a	63.7 ± 7.4 a	79.5 ± 10.2 a	70.5 ± 4.4 a	57.9 ± 3.6 a	115.3 ± 7.2 b
Trihydroxytetralone galloyl hexoside	68.6 ± 8.7 a	95.3 ± 15.5 a	86.5 ± 6.0 a	73.4 ± 8.6 a	90.2 ± 9.7 a	106.7 ± 12.7 a
**Hydroxycinnamic acids**						
Neochlorogenic acid (3-caffeoylquinic acid)	12.2 ± 0.7 a	17.5 ± 1.7 a	16.4 ± 2.1 a	14.3 ± 1.1 a	16.1 ± 1.4 a	18.0 ± 1.3 a
3-*p*-Coumaroylquinic acid	66.5 ± 3.6 ab	101.7 ± 5.5 c	104.9 ± 4.0 c	70.1 ± 3.0 ab	60.4 ± 2.3 a	82.1 ± 2.9 b
*p*-Coumaric acid derivative 2	4.5 ± 0.5 ab	4.0 ± 0.3 ab	3.8 ± 0.4 a	6.7 ± 1.0 ac	7.8 ± 0.8 c	6.9 ± 0.6 bc
*p*-Coumaric acid derivative 3	9.1 ± 0.6 b	6.2 ± 0.3 ab	7.2 ± 0.7 ab	6.0 ± 0.2 a	14.2 ± 1.0 c	15.3 ± 0.8 c
*p*-Coumaric acid derivative 4	4.9 ± 0.6 a	5.0 ± 0.3 a	5.3 ± 0.5 a	5.4 ± 0.4 a	6.1 ± 0.3 a	6.5 ± 0.4 a
Caffeic acid derivative 4	2.5 ± 0.9 a	3.2 ± 0.2 ab	3.1 ± 0.4 ab	7.7 ± 0.8 c	6.9 ± 1.4 bc	9.7 ± 1.0 c
Ferulic acid derivative 2	15.3 ± 1.0 b	20.2 ± 1.3 c	21.8 ± 1.6 c	15.1 ± 0.5 b	9.9 ± 0.7 a	15.1 ± 0.6 b
Ferulic acid derivative 3	6.3 ± 0.2 a	8.1 ± 0.8 ab	11.8 ± 1.0 bc	8.7 ± 1.1 ab	13.6 ± 1.0 c	14.8 ± 1.2 c
**Hydroxybenzoic acids**						
Gallic acid derivative 3	142.2 ± 14.1 a	85.9 ± 10.6 a	132.2 ± 3.5 a	244.6 ± 18.6 b	331.3 ± 16.1 c	366.9 ± 11.8 c
Gallic acid derivative 4	16.7 ± 2.7 ab	17.6 ± 2.7 ab	23.6 ± 4.2 ab	13.0 ± 1.7 a	16.5 ± 2.6 ab	25.4 ± 1.7 b
Gallic acid derivative 5	12.9 ± 2.4 a	13.6 ± 2.2 a	14.1 ± 2.8 a	15.6 ± 1.6 a	25.7 ± 2.0 b	17.5 ± 1.3 ab
**Flavanols**						
(+)Catechin	53.7 ± 2.9 ab	82.2 ± 4.4 c	84.8 ± 3.2 c	56.7 ± 2.4 ab	48.9 ± 1.8 a	66.4 ± 2.4 b
(−)Epicatechin	69.4 ± 5.0 a	56.3 ± 5.0 a	65.7 ± 3.9 a	54.8 ± 4.0 a	105.0 ± 5.2 b	139.3 ± 9.4 c
(epi)Catechin derivative 5	50.6 ± 8.9 a	56.9 ± 8.2 ab	68.8 ± 3.3 ac	79.4 ± 4.7 bc	88.0 ± 0.8 c	80.9 ± 1.6 bc
**Flavones**						
Santin	14.1 ± 1.6 a	13.3 ± 1.9 a	15.0 ± 1.9 a	35.6 ± 4.8 b	49.4 ± 1.1 c	46.2 ± 3.7 bc
5,7-Dihydroxy-3,4-dimetoxyflavone	27.1 ± 0.4 bc	12.5 ± 1.8 a	16.2 ± 2.8 ab	20.3 ± 2.0 ab	40.5 ± 4.4 c	39.8 ± 4.4 c
**Flavonols**						
Quercetin-3-galactoside	33.6 ± 1.8 ab	39.8 ± 2.5 ab	34.7 ± 2.1 ab	42.3 ± 3.1 b	35.8 ± 1.8 ab	31.4 ± 0.6 a
Quercetin-3-rhamnoside	12.7 ± 3.1 a	22.7 ± 3.5 a	20.6 ± 1.5 a	16.8 ± 2.4 a	17.4 ± 0.4 a	16.3 ± 1.2 a
**Total naphthoquinones**	2946.7 ± 116.4 a	3026.2 ± 133.4 a	3202.6 ± 142.9 a	4454.6 ± 235.0 b	4670.0 ± 74.7 b	5088.8 ± 156.9 b
**Total hydroxycinnamic acids**	121.2 ± 4.3 a	166.0 ± 7.7 b	174.3 ± 8.8 b	134.0 ± 4.4 a	135.0 ± 6.3 a	168.4 ± 5.0 b
**Total hydroxybenzoic acids**	171.8 ± 17.9 a	117.1 ± 14.0 a	169.9 ± 9.4 a	273.2 ± 20.0 b	373.5 ± 17.3 c	409.8 ± 13.4 c
**Total flavanols**	173.7 ± 11.3 a	195.5 ± 15.3 a	219.4 ± 8.7 ab	190.9 ± 8.3 a	241.9 ± 7.2 bc	286.6 ± 8.9 c
**Total flavones**	41.1 ± 1.8 ab	25.7 ± 3.2 a	31.2 ± 4.0 a	55.9 ± 6.7 b	89.9 ± 5.0 c	86.0 ± 7.3 c
**Total flavanones**	nd	nd	nd	nd	Nd	nd
**Total flavonols**	46.3 ± 4.1 a	62.4 ± 5.2 a	55.2 ± 3.4 a	59.1 ± 4.8 a	53.2 ± 2.0 a	47.8 ± 1.7 a
**Total phenolics content (summation; relevant standards) ^x^**	3500.8 ± 137.3 a	3592.8 ± 155.3 a	3852.7 ± 173.1 a	5167.7 ± 275.4 b	5563.5 ± 81.0 bc	6087.3 ± 179.6 c
**Total phenolics content (total extracts; mg gallic acid equivalents/100 g dry weight) ^y^**	1156.6 ± 51.4 a	1532.7 ± 105.6 b	1398.9 ± 80.2 ab	1241.2 ± 56.6 ab	1155.2 ± 22.9 a	1398.5 ± 102.8 ab

Data are means ± standard error. ^x^ expressed as the sum of all of the individual identified phenolics (summation), in mg/100 g dry weight of the most relevant standard. ^y^ expressed as the separate analysis of the total phenolics for each extract (total extracts), in mg gallic acid equivalents/100 g dry weight. Means followed by different letters within a cultivar are significantly different (*p* ≤ 0.05; Tukey’s tests); nd, not detected.

**Table 6 biology-10-00535-t006:** Individual phenolics for the walnut buds across the six selected cultivars.

Compound	Bud Phenolic Content per Cultivar (mg/100 g Dry Weight)					
	‘Fernor’	‘Fernette’	‘Franquette’	‘Sava’	‘Krka’	‘Rubina’
**Naphthoquinones**						
Juglone	573.8 ± 30.3 c	407.5 ± 24.1 ab	373.6 ± 9.9 a	392.3 ± 5.9 a	466.0 ± 18.2 ab	508.8 ± 29.1 bc
Hydrojuglone	149.4 ± 6.7 c	100.4 ± 2.9 ab	113.4 ± 10.7 bc	70.7 ± 11.8 a	88.0 ± 7.0 ab	111.6 ± 9.0 bc
Hydrojuglone β-D-glucopyranoside	2744.4 ± 58.1 c	1636.6 ± 11.4 a	3619.9 ± 100.3 d	2326.5 ± 25.5 b	2148.8 ± 37.8 b	1688.2 ± 6.4 a
Hydrojuglone rutinoside	314.7 ± 14.4 c	191.1 ± 4.0 a	283.0 ± 13.9 c	189.4 ± 11.9 a	261.1 ± 13.1 bc	222.5 ± 6.7 ab
Hydrojuglone dihexoside	509.0 ± 17.0 b	404.3 ± 14.8 a	683.1 ± 20.3 d	606.5 ± 12.0 c	454.7 ± 13.7 ab	444.2 ± 14.0 ab
Hydrojuglone derivative 1	642.3 ± 27.3 b	348.1 ± 18.7 a	951.0 ± 27.1 c	432.1 ± 15.4a	440.7 ± 8.5 a	432.5 ± 16.1 a
Hydrojuglone derivative 2	109.1 ± 6.2 a	57.1 ± 3.6 a	868.7 ± 14.7 e	270.9 ± 7.2 b	526.8 ± 17.8 c	594.1 ± 12.6 d
Hydrojuglone derivative 3	301.2 ± 11.5 c	165.3 ± 8.6 a	260.3 ± 17.5 bc	161.0 ± 8.4 a	180.7 ± 11.3 a	216.1 ± 10.8 ab
Hydrojuglone derivative pentoside 1	1038.7 ± 18.2 d	594.0 ± 18.9 a	1995.1 ± 30.9 e	973.0 ± 11.6 cd	824.0 ± 17.6 b	916.5 ± 27.1 bc
Hydrojuglone derivative pentoside 2	3855.8 ± 86.1 d	2138.6 ± 29.4 ab	3021.7 ± 34.9 c	3166.0 ± 50.0 c	2370.3 ± 46.4 b	1940.6 ± 34.8 a
Hydrojuglone derivative rhamnoside	1525.3 ± 31.7 c	861.5 ± 18.3 a	2429.3 ± 36.9 e	1297.8 ± 33.8 b	1933.2 ± 29.1 d	1646.5 ± 14.1 c
Hydrojuglone pentose galloyl derivative	433.9 ± 25.8 c	234.5 ± 5.6 a	525.5 ± 14.7 d	268.0 ± 11.9 ab	288.8 ± 24.0 ab	341.7 ± 12.8 b
Dihydroxytetralone hexoside	285.0 ± 7.8 c	194.1 ± 13.7 a	382.4 ± 13.1 d	281.8 ± 9.0 bc	219.5 ± 14.6 ab	257.2 ± 20.0 ac
**Hydroxycinnamic acids**						
Neochlorogenic acid (3-caffeoylquinic acid)	74.2 ± 0.9 c	29.2 ± 2.6 a	99.4 ± 6.8 d	58.3 ± 3.1 bc	62.0 ± 3.9 bc	51.2 ± 4.3 b
3-*p*-Coumaoylquinic acid	323.1 ± 5.2 c	165.9 ± 4.7 a	395.3 ± 10.4 d	273.4 ± 4.6 b	257.5 ± 2.2 b	283.3 ± 4.3 b
*p*-Coumaric acid derivative 1	70.8 ± 0.9 c	45.7 ± 0.8 a	100.3 ± 1.5 d	62.1 ± 0.9 b	65.8 ± 0.6 b	42.1 ± 0.4 a
Caffeic acid hexoside derivative	20.5 ± 0.7 a	18.5 ± 0.6 a	48.5 ± 2.2 b	22.2 ± 1.8 a	18.9 ± 0.7 a	23.0 ± 0.6 a
Caffeic acid derivative 1	9.0 ± 0.7 b	6.5 ± 0.3 a	12.0 ± 0.4 c	4.9 ± 0.6 a	6.8 ± 0.5 ab	6.8 ± 0.4 ab
Diferuoyl hexoside	8.4 ± 0.5 c	5.6 ± 0.3 b	8.0 ± 0.5 c	2.8 ± 0.1 a	7.1 ± 0.4 bc	7.3 ± 0.3 bc
**Hydroxybenzoic acids**						
Gallic acid derivative 1	55.2 ± 1.5 b	37.4 ± 1.5 a	60.7 ± 1.0 b	41.9 ± 1.5 a	42.3 ± 1.2 a	42.2 ± 1.1 a
Gallic acid derivative 2	705.9 ± 4.3 b	369.8 ± 157.6 a	1209.4 ± 26.3 c	811.7 ± 22.3 b	655.2 ± 6.3 ab	685.1 ± 7.2 b
Gallic acid derivative 3	541.6 ± 6.7 d	305.1 ± 6.7 a	424.2 ± 9.9 b	485.5 ± 6.1 c	520.6 ± 7.1 cd	416.0 ± 9.5 b
Gallic acid methyl ester	90.9 ± 1.6 b	62.1 ± 5.7 a	188.9 ± 9.5 c	92.1 ± 4.5 b	92.1 ± 4.2 b	95.7 ± 2.3 b
*bis*-HHDP-glucose	70.7 ± 3.7 bc	46.1 ± 2.0 a	80.8 ± 4.9 c	72.9 ± 2.3 bc	41.8 ± 1.3 a	63.6 ± 0.7 b
Ellagic acid derivative	451.4 ± 3.9 d	226.0 ± 13.5 a	942.8 ± 17.0 e	366.7 ± 7.1 c	276.5 ± 3.6 b	333.1 ± 10.4 c
**Flavanols**						
Procyanidin dimer 1	303.8 ± 18.4 b	195.4 ± 21.8 a	256.1 ± 11.5 ab	289.7 ± 3.5 b	275.9 ± 5.8 b	267.1 ± 10.6 b
Procyanidin dimer 2	494.3 ± 9.8 cd	350.9 ± 25.4 a	527.6 ± 16.2 d	507.1 ± 17.2 cd	378.9 ± 19.6 ab	433.9 ± 7.9 bc
Procyanidin dimer derivative 1	655.6 ± 24.0 c	293.9 ± 10.1 a	859.0 ± 40.5 d	604.1 ± 11.7 c	589.6 ± 2.7 c	452,4 ± 10.8 b
Procyanidin dimer derivative 2	431.7 ± 13.9 b	261.6 ± 3.4 a	823.9 ± 18.2 d	445.2 ± 15.1 b	413.3 ± 7.6 b	630.5 ± 22.8 c
Procyanidin dimer derivative 3	122.4 ± 7.3 c	88.0 ± 5.6 ab	193.8 ± 8.3 d	72.4 ± 3.7 a	116.1 ± 3.6 bc	116.6 ± 7.9 bc
(+)Catechin	838.6 ± 34.4 c	469.0 ± 6.6 a	1218.0 ± 25.9 d	666.2 ± 22.9 b	664.8 ± 16.2 b	744.9 ± 26.5 bc
(−)Epicatechin	266.6 ± 6.0 c	178.3 ± 4.2 a	347.4 ± 7.0 d	198.5 ± 5.2 a	238.5 ± 8.1 b	192.8 ± 1.2 a
(epi)Catechin derivative 1	210.9 ± 2.7 b	160.5 ± 5.0 a	273.1 ± 4.8 c	205.8 ± 1.9 b	205.5 ± 5.3 b	209.8 ± 2.5 b
(epi)Catechin derivative 2	267.3 ± 7.1 c	185.1 ± 15.7 a	384.7 ± 10.0 d	269.3 ± 13.2 c	256.4 ± 13.4 bc	200.6 ± 19.6 ab
(epi)Catechin derivative 3	435.9 ± 9.9 c	291.5 ± 6.8 a	568.0 ± 11.4 d	324.5 ± 8.5 a	390.0 ± 13.3 b	315.3 ± 2.0 a
(epi)Catechin derivative 4	94.7 ± 4.1 a	80.4 ± 4.5 a	155.5 ± 4.4 b	91.3 ± 3.2 a	82.3 ± 3.9 a	92.8 ± 2.9 a
Galloyl-3-(epi)catechin	850.0 ± 9.3 d	461.8 ± 15.3 a	1396.0 ± 23.6 e	644.6 ± 20.7 b	724.0 ± 20.0 bc	797.5 ± 6.3 cd
**Flavones**						
Santin	45.9 ± 2.3 c	35.3 ± 1.7 ab	26.6 ± 2.3 a	32.2 ± 2.0 ab	28.3 ± 1.6 ab	36.8 ± 1.6 bc
5,7-Dihydroxy-3,4-dimetoxyflavone	45.3 ± 2.3 c	38.6 ± 2.6 bc	28.5 ± 1.8 a	29.2 ± 2.5 ab	35.9 ± 1.7 ac	58.4 ± 1.0 d
**Flavanones**						
Naringenin	83.5 ± 3.5 ab	65.7 ± 4.7 a	83.9 ± 4.7 ab	73.9 ± 2.7 a	70.3 ± 4.8 a	98.4 ± 3.3 b
**Flavonols**						
Myricetin galactoside	259.0 ± 1.4 d	140.2 ± 3.9 b	214.0 ± 8.0 c	141.4 ± 3.2 b	131.5 ± 5.1 b	107.7 ± 3.0 a
Myricetin pentoside	105.4 ± 4.8 b	54.4 ± 2.1 a	242.3 ± 8.5 c	92.4 ± 4.7 b	95.6 ± 4.1 b	104.8 ± 6.0 b
Myricetin-3-rhamnoside	533.3 ± 12.2 c	356.6 ± 2.1 a	770.6 ± 14.9 d	553.9 ± 9.9 c	452.2 ± 12.6 b	365.0 ± 7.7 a
Quercetin-3-galactoside	227.5 ± 2.5 d	123.6 ± 4.1 a	373.6 ± 6.3 e	172.5 ± 5.5 b	193.8 ± 5.4 bc	213.4 ± 1.7 cd
Quercetin-3-glucoside	144.0 ± 4.3 bc	113.9 ± 4.6 a	262.0 ± 6.7 d	147.9 ± 6.5 bc	158.4 ± 4.1 c	133.6 ± 3.4 ab
Quercetin-3-arabinopyranoside	293.4 ± 3.6 b	206.9 ± 3.1 a	495.9 ± 9.8 c	287.9 ± 6.5 b	308.9 ± 2.5 b	282.5 ± 5.3 b
Quercetin-3-arabinofuranoside	250.8 ± 12.6 a	207.2 ± 2.8 a	613.0 ± 19.3 c	244.3 ± 8.9 a	335.3 ± 6.0 b	320.2 ± 5.8 b
Quercetin-3-rhamnoside	399.1 ± 7.6 bc	332.5 ± 4.9 a	745.4 ± 25.7 d	371.0 ± 7.2 ab	440.7 ± 8.5 c	424.7 ± 7.6 bc
Quercetin galoyll hexoside	130.8 ± 4.4 c	90.5 ± 2.3 a	177.5 ± 4.6 d	143.7 ± 1.6 c	98.4 ± 2.9 ab	107.2 ± 2.5 b
Quercetin hexoside derivative 1	54.5 ± 2.8 c	34.4 ± 1.6 ab	68.6 ± 4.3 d	27.2 ± 1.4 a	50.6 ± 1.2 c	41.8 ± 3.4 bc
Quercetin hexoside derivative 2	36.3 ± 1.0 bc	30.4 ± 1.6 b	40.2 ± 0.9 c	19.6 ± 1.3 a	30.7 ± 1.5 b	31.5 ± 1.5 b
Quercetin	29.0 ± 1.1 c	21.4 ± 0.9 b	29.6 ± 1.4 c	15.6 ± 1.3 a	21.3 ± 0.8 b	26.8 ± 1.3 bc
Kaempferol pentoside 1	39.1 ± 1.0 b	22.9 ± 1.9 a	76.6 ± 4.9 c	26.7 ± 0.9 a	29.5 ± 2.1 ab	40.2 ± 1.4 b
Kaempferol pentoside 2	68.8 ± 2.9 b	43.1 ± 0.9 a	114.6 ± 4.7 c	61.0 ± 3.0 b	56.5 ± 2.3 ab	53.4 ± 4.4 ab
Kaempferol pentoside 3	36.2 ± 1.6 c	17.9 ± 0.4 a	66.5 ± 2.2 d	28.4 ± 1.7 bc	29.1 ± 2.0 bc	27.8 ± 1.9 b
Kaempferol rhamnoside	53.0 ± 1.4 bc	37.5 ± 3.9 a	83.4 ± 3.6 d	45.0 ± 4.2 ab	46.1 ± 2.5 ab	60.9 ± 2.0 c
Kaempferol	21.3 ± 1.2 a	20.8 ± 1.6 a	18.5 ± 1.7 a	20.6 ± 1.7 a	20.9 ± 1.5 a	25.1 ± 1.4 a
**Total naphthoquinones**	12482.7 ± 126.9 d	7333.0 ± 102.7 a	15507.1 ± 122.8 e	10435.9 ± 42.1 c	10202.6 ± 31.8 c	9320.5 ± 91.8 b
**Total hydroxycinnamic acids**	506.1 ± 4.9 c	271.2 ± 7.9 a	663.4 ± 17.7 d	423.7 ± 10.1 b	418.1 ± 2.0 b	413.7 ± 3.2 b
**Total hydroxybenzoic acids**	1915.8 ± 2.1 b	1046.6 ± 156.6 a	2906.8 ± 46.2 c	1870.8 ± 21.0 b	1628.5 ± 20.1 b	1635.7 ± 10.9 b
**Total flavanols**	4971.9 ± 61.0 c	3016.5 ± 43.1 a	7003.2 ± 129.6 d	4318.6 ± 9.9 b	4335.3 ± 59.0 b	4454.1 ± 32.6 b
**Total flavones**	91.3 ± 4.5 c	73.9 ± 4.0 b	55.1 ± 4.0 a	61.4 ± 0.6 ab	64.2 ± 1.2 ab	95.3 ± 0.7 c
**Total flavanones**	83.5 ± 3.5 ab	65.7 ± 4.7 a	83.9 ± 4.7 ab	73.9 ± 2.7 a	70.3 ± 4.8 a	98.4 ± 3.3 b
**Total flavonols**	2681.4 ± 7.6 d	1854.1 ± 16.2 a	4392.3 ± 42.3 e	2399.0 ± 19.6 bc	2499.3 ± 18.8 c	2366.7 ± 13.8 b
**Total phenolics content (summation; relevant standards) ^x^**	22732.6 ± 189.8 d	13661.1 ± 283.3 a	30611.8 ± 130.3 e	19583.4 ± 74.3 c	19218.3 ± 21.1 c	18384.4 ± 142.3 b
**Total phenolics content (total extracts; mg gallic acid equivalents/100 g dry weight) ^y^**	7236.4 ± 188.1 c	4270.2 ± 144.4 a	8232.8 ± 57.7 d	5199.6 ± 166.3 b	5017.7 ± 220.7 ab	4853.8 ± 259.2 ab

Data are means ± standard error. ^x^ expressed as the sum of all of the individual identified phenolics (summation), in mg/100 g dry weight of the most relevant standard. ^y^ expressed as the separate analysis of the total phenolics for each extract (total extracts), in mg gallic acid equivalents/100 g dry weight. Means followed by different letters within a cultivar are significantly different (*p* ≤ 0.05; Tukey’s tests); nd, not detected.

**Table 7 biology-10-00535-t007:** Individual phenolics for the walnut bark across the six selected cultivars.

Compound	Bark Phenolic Content per Cultivar (mg/100 g Dry Weight)					
	‘Fernor’	‘Fernette’	‘Franquette’	‘Sava’	‘Krka’	‘Rubina’
**Naphthoquinones**						
Juglone	251.0 ± 31.2 ac	185.8 ± 8.6 ab	281.2 ± 16.6 bc	169.4 ± 10.2 a	219.4 ± 24.1 ac	299.8 ± 35.6 c
Hydrojuglone	72.8 ± 2.2 bc	89.9 ± 9.6 c	50.7 ± 7.2 ab	65.9 ± 5.5 bc	21.3 ± 2.4 a	66.9 ± 9.8 bc
Hydrojuglone β-D-glucopyranoside	239.8 ± 13.0 a	366.1 ± 18.7 b	342.3 ± 17.8 ab	285.5 ± 26.2 ab	255.1 ± 11.9 ab	523.4 ± 52.7 c
Hydrojuglone rutinoside	47.6 ± 4.8 ab	30.9 ± 3.1 a	40.6 ± 5.5 a	34.9 ± 2.8 a	45.1 ± 3.8 ab	60.9 ± 2.1 b
Hydrojuglone derivative 2	16.6 ± 5.1 a	10.5 ± 4.7 a	156.3 ± 21.1 b	32.0 ± 9.2 a	143.9 ± 9.4 b	143.3 ± 22.0 b
Hydrojuglone derivative 4	1258.0 ± 73.5 c	877.8 ± 53.9 b	639.1 ± 83.0 ab	471.6 ± 35.5 a	417.4 ± 17.0 a	535.2 ± 78.9 a
Hydrojuglone derivative pentoside 1	566.3 ± 25.2 b	757.7 ± 34.4 c	401.4 ± 14.1 a	699.1 ± 35.5 bc	363.2 ± 21.9 a	394.7 ± 42.7 a
Hydrojuglone derivative pentoside 2	6579.7 ± 402.1 b	7608.0 ± 301.1 b	4768.3 ± 337.3 a	7189.0 ± 266.6 b	4672.8 ± 174.5 a	4954.1 ± 494.4 a
Hydrojuglone derivative pentoside 3	605.9 ± 24.3 bc	743.9 ± 32.8 c	487.3 ± 24.5 ab	681.9 ± 36.8 c	427.7 ± 16.0 a	460.3 ± 49.8 ab
Hydrojuglone derivative rhamnoside	2015.2 ± 110.6 a	2332.2 ± 103.3 a	2172.8 ± 157.9 a	2039.4 ± 75.6 a	2222.2 ± 123.3 a	2230.2 ± 193.9 a
Dihydroxytetralone hexoside	43.4 ± 3.0 a	54.0 ± 4.7 a	48.8 ± 2.8 a	47.2 ± 6.8 a	37.7 ± 5.7 a	49.6 ± 3.5 a
**Hydroxybenzoic acids**						
Gallic acid derivative 2	80.0 ± 5.3 ab	89.9 ± 5.3 ab	106.7 ± 4.9 b	92.5 ± 7.0 ab	71.8 ± 3.5 a	93.0 ± 8.9 ab
Gallic acid derivative 3	29.7 ± 3.1 bc	31.1 ± 1.5 c	15.6 ± 0.8 a	35.2 ± 3.6 cd	16.8 ± 2.3 ab	44.1 ± 4.3 d
Ellagic acid derivative	145.6 ± 12.6 b	153.9 ± 8.9 b	155.8 ± 9.7 b	93.2 ± 9.4 a	111.0 ± 7.1 ab	123.0 ± 13.6 ab
**Flavanols**						
Procyanidin dimer 2	219.8 ± 18.8 a	277.1 ± 20.2 a	301.3 ± 25.6 a	335.3 ± 24.5 a	288.7 ± 14.6 a	333.2 ± 45.3 a
Procyanidin dimer derivative 2	200.9 ± 10.5 ab	248.2 ± 21.8 b	213.0 ± 17.1 ab	245.4 ± 17.8 b	149.8 ± 7.8 a	178.4 ± 25.6 ab
(+)Catechin	528.2 ± 56.6 a	557.9 ± 40.9 a	675.5 ± 42.0 a	708.7 ± 49.5 a	748.8 ± 53.0 a	797.2 ± 103.9 a
**Flavones**						
Santin	14.4 ± 2.0 b	6.3 ± 0.4 a	7.7 ± 0.6 a	10.6 ± 2.2 ab	6.6 ± 0.3 a	7.8 ± 0.1 a
5,7-Dihydroxy-3,4-dimetoxyflavone	2.2 ± 0.6 ab	0.9 ± 0.1 a	4.3 ± 0.3 b	4.4 ± 0.9 b	2.4 ± 0.2 ab	3.6 ± 0.5 b
**Flavonols**						
Myricetin pentoside	21.4 ± 1.3 a	26.9 ± 1.5 a	29.4 ± 1.0 a	29.1 ± 2.7 a	23.1 ± 3.2 a	23.1 ± 3.2 a
Myricetin-3-rhamnoside	129.4 ± 9.0 a	156.3 ± 8.2 a	162.7 ± 3.9 a	152.1 ± 12.5 a	121.0 ± 6.8 a	133.4 ± 12.9 a
Quercetin-3-galactoside	210.4 ± 10.7 a	261.9 ± 19.1 ab	248.4 ± 9.7 ab	322.1 ± 28.4 b	224.9 ± 6.5 a	228.0 ± 25.0 a
Quercetin-3-glucoside	141.0 ± 6.2 ab	159.0 ± 5.1 ab	124.3 ± 11.1 ab	165.4 ± 9.2 b	115.7 ± 4.2 a	133.3 ± 17.6 ab
Quercetin-3-arabinopyranoside	69.0 ± 2.3 a	93.5 ± 3.6 ab	91.5 ± 6.1 ab	100.1 ± 8.4 b	74.1 ± 1.8 ab	78.1 ± 9.7 ab
Quercetin-3-arabinofuranoside	69.6 ± 1.9 ab	100.4 ± 7.4 bc	96.7 ± 7.9 ac	123.4 ± 6.7 c	64.9 ± 2.4 a	92.4 ± 14.0 ac
Quercetin-3-rhamnoside	223.8 ± 11.5 ab	320.1 ± 13.3 bc	255.5 ± 33.4 ab	373.9 ± 19.0 c	171.7 ± 5.5 a	315.1 ± 47.4 bc
Quercetin galoyll hexoside	82.7 ± 5.8 ab	97.3 ± 4.9 ab	82.6 ± 7.0 ab	101.6 ± 6.5 b	69.5 ± 3.0 a	76.9 ± 10.0 ab
Kaempferol-7-hexoside 1	150.5 ± 9.7 bc	237.1 ± 8.6 d	113.0 ± 18.3 ab	195.3 ± 11.7 cd	63.5 ± 6.3 a	107.7 ± 23.5 ab
Kaempferol-7-hexoside 2	30.2 ± 1.3 b	42.2 ± 1.5 c	32.2 ± 3.6 bc	26.6 ± 2.4 b	14.8 ± 2.3 a	34.1 ± 2.7 bc
**Total naphthoquinones**	11696.3 ± 654.3 bc	13056.9 ± 527.9 c	9388.8 ± 596.2 ab	11716.0 ± 488.3 bc	8825.9 ± 339.7 a	9718.4 ± 919.8 ab
**Total hydroxycinnamic acids**	nd	nd	nd	nd	Nd	nd
**Total hydroxybenzoic acids**	255.3 ± 15.9 a	274.9 ± 15.4 a	278.2 ± 14.1 a	220.8 ± 19.8 a	199.6 ± 11.4 a	260.2 ± 25.9 a
**Total flavanols**	948.9 ± 84.2 a	1083.3 ± 78.6 a	1189.8 ± 81.3 a	1289.3 ± 88.6 a	1187.3 ± 72.3 a	1308.7 ± 174.5 a
**Total flavones**	16.6 ± 2.0 c	7.2 ± 0.4 a	12.0 ± 0.8 ac	15.0 ± 2.9 bc	9.0 ± 0.5 ab	11.4 ± 0.6 ac
**Total flavanones**	nd	nd	nd	nd	Nd	nd
**Total flavonols**	1128.0 ± 17.3 ab	1494.5 ± 63.9 bc	1236.2 ± 97.9 ac	1589.5 ± 89.0 c	943.2 ± 16.5 a	1222.2 ± 160.8 ac
**Total phenolics content (summation; relevant stadnards) ^x^**	14045.2 ± 757.1 ac	15916.8 ± 676.0 c	12104.9 ± 774.8 ab	14830.6 ± 673.2 bc	11165.1 ± 422.7 a	12520.9 ± 1270.9 ac
**Total phenolics content (total extracts; mg gallic acid equivalents/100 g dry weight) ^y^**	1956.2 ± 77.6 a	2236.4 ± 251.1 a	2544.4 ± 251.3 a	1898.2 ± 121.7 a	1979.3 ± 38.7 a	2740.0 ± 440.6 a

Data are means ± standard error. ^x^ expressed as the sum of all of the individual identified phenolics (summation), in mg/100 g dry weight of the most relevant standard. ^y^ expressed as the separate analysis of the total phenolics for each extract (total extracts), in mg gallic acid equivalents/100 g dry weight. Means followed by different letters within a cultivar are significantly different (*p* ≤ 0.05; Tukey’s tests); nd, not detected.

**Table 8 biology-10-00535-t008:** Overview of the 83 phenolics identified for the walnut inner and outer husks, buds and bark.

Phenolic	Husk		Buds	Bark
	Inner	Outer		
**Naphthoquinones**				
1,4-Naphthoquinone	+	+		
Juglone	+	+	+	+
Hydrojuglone	+	+	+	+
Hydrojuglone β-D-glucopyranoside	+	+	+	+
Hydrojuglone rutinoside	+	+	+	+
Hydrojuglone dihexoside			+	
Hydrojuglone derivative 1			+	
Hydrojuglone derivative 2			+	+
Hydrojuglone derivative 3			+	
Hydrojuglone derivative 4				+
Hydrojuglone derivative 5	+	+		
Hydrojuglone derivative pentoside 1			+	+
Hydrojuglone derivative pentoside 2	+	+	+	+
Hydrojuglone derivative pentoside 3				+
Hydrojuglone derivative rhamnoside	+	+	+	+
Hydrojuglone pentose galloyl derivative			+	
Hydrojuglone hexoside derivative	+	+		
*bis*-Juglone	+	+		
*p*-Hydroxymetoxybenzobijuglone	+	+		
Regiolone	+	+		
5-Hydroxy-2,3-dihydro-1,4-naphthalenedione		+		
4,5,8-Trihydroxynaphthalene-5-D-glucopyranoside	+	+		
1,4,8-Trihydroxynaphthalene-1-D-glucopyranoside	+	+		
Dihydroxytetralone hexoside	+	+	+	+
Dihydroxytetralone galloyl hexoside	+	+		
**Hydroxycinnamic acids**				
Neochlorogenic acid (3-caffeoylquinic acid)	+	+	+	
3-*p*-Coumaroylquinic acid	+	+	+	
*p*-Coumaric acid derivative 1			+	
*p*-Coumaric acid derivative 2	+	+		
*p*-Coumaric acid derivative 3	+	+		
*p*-Coumaric acid derivative 4	+	+		
Caffeic acid hexoside derivative			+	
Caffeic acid derivative 1			+	
Caffeic acid derivative 2	+			
Caffeic acid derivative 3	+			
Caffeic acid derivative 4	+	+		
Diferuoyl hexoside			+	
Ferulic acid derivative 1	+			
Ferulic acid derivative 2	+	+		
Ferulic acid derivative 3	+	+		
**Hydroxybenzoic acids**				
Gallic acid derivative 1			+	
Gallic acid derivative 2			+	+
Gallic acid derivative 3	+	+	+	+
Gallic acid derivative 4	+	+		
Gallic acid derivative 5	+	+		
Gallic acid methyl ester			+	
*bis*-HHDP-glucose			+	
Ellagic acid derivative			+	+
**Flavanols**				
Procyanidin dimer 1			+	
Procyanidin dimer 2			+	+
Procyanidin dimer derivative 1			+	
Procyanidin dimer derivative 2			+	+
Procyanidin dimer derivative 3			+	
(+)Catechin	+	+	+	+
(−)Epicatechin	+	+	+	
(epi)Catechin derivative 1			+	
(epi)Catechin derivative 2			+	
(epi)Catechin derivative 3			+	
(epi)Catechin derivative 4			+	
(epi)Catechin derivative 5		+		
Galloyl-3-(epi)catechin			+	
**Flavones**				
Santin	+	+	+	+
5,7-Dihydroxy-3,4-dimetoxyflavone	+	+	+	+
**Flavanones**				
Naringenin			+	
**Flavonols**				
Myricetin galactoside			+	
Myricetin pentoside			+	+
Myricetin-3-rhamnoside			+	+
Quercetin-3-galactoside	+	+	+	+
Quercetin-3-glucoside			+	+
Quercetin-3-arabinopyranoside			+	+
Quercetin-3-arabinofuranoside			+	+
Quercetin-3-rhamnoside	+	+	+	+
Quercetin galoyll hexoside			+	+
Quercetin hexoside derivative 1			+	
Quercetin hexoside derivative 2			+	
Quercetin			+	
Kaempferol pentoside 1			+	
Kaempferol pentoside 2			+	
Kaempferol pentoside 3			+	
Kaempferol rhamnoside			+	
Kaempferol-7-hexoside 1				+
Kaempferol-7-hexoside 2				+
Kaempferol			+	

+, phenolic identified.

## Data Availability

Part of the data presented in this study are available in Appendix A here. The remaining data presented in this study are available on request from the corresponding author. The remaining data are not publicly available due to privacy.

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
