# Peer review of "Walnut (J. regia) Agro-Residues as a Rich Source of Phenolic Compounds"

_biology, 2021, doi:10.3390/biology10060535_

Round 1
Reviewer 1 Report
This study identified and quantified phenolic compounds in the inner and outer husks, buds, and bark of walnuts. The authors should be commended for their work in identifying these phenolics. However, it is unclear how this information can be applied.
They mentioned these phenolics can be used for the following: to “preserve the environment”, boost the “economic outcome for farmers and companies”, as a “sorbent for oil”, for “hazardous material removal”, as “an ingredient in the cosmetics industry”, as a “blasting medium”, “as a biofuel”, “as a natural dye”, and to develop “new and valuable drugs”. They also stated that the identification and quantification of phenolics in walnut agro-residues “might help to propose new directions for further studies essen[t]ial for agro-food, cosmet[i]cs and farmacy industries." However, these statements are vague, and thus, require further details and clarifications. As indicated, there are also spelling errors. Furthermore, the authors discussed some of the human health benefits of bioactive compounds. However, other plants and food items contain many of these phenolics. How are walnuts unique in this aspect? This should be emphasized. They also investigated the inedible portions of the walnut, which would not be applicable here. However, they did mention that these isolated walnut phenolics could be used as functional ingredients in foods. Again, how are these phenolics unique to walnuts and what are the specific benefits and uses?
Author Response
Kind regards,
We thank you for your time and review and the concerns you have expressed regarding the manuscript. All responses can be found in the attached file.
Kind regards

Reviewer 2 Report
Peer review report on “Walnut (J. regia) agro-residues as a rich source of phenolic compounds.”
Manuscript ID: biology-1252153
The value of discarded residues from agriculture is being investigated more frequently as sustainability in agricultural systems is recognized as an important part of agriculture in the 21st century. Therefore, this paper on the identification and quantification of 83 major phenolic compounds to be found in the inner and outer husks, buds and bark of the Persian walnut, and comparisons across six different cultivars presents a huge amount of useful data on the by-products of the walnut industry.
This paper represents a large and detailed body of work. It is well written, described and presented. The analytical techniques used included UHPLC and MS/MS analysis based on the use of standard compounds and literature examples and appear to be comprehensive, thorough, and well carried out.
Some comments:
Tables 1, 2 and 3: Obviously space is at a premium, but Entry numbers would be useful for the listed phenolics.
Table 1:
Entry 6: should be 3-p-Coumaroylquinic acid.
Entry 20: (epi)Catechin derivative 5. In Figure 4 (Supplementary info), the m/z looks more like 470 than 469. Would you clarify please.
The fragmentation pattern for MS3 in the table appears to have been omitted. Figure 5 (Supplementary info) exhibits the typical pattern for MS3 of 245, 205, 179, 125 as seen for other (epi)Catechin derivatives.
Entry 26: Quercitin-3-rhamnoside. The m/z says 447 but it is not visible on the spectra in Figure 8 (Supplementary info). Please clarify.
Table 2:
Entry 9: Should be 3-p-Coumaroyalquinic acid.
Tables 4, 5, 6. 8. Under Hydroxycinnamic acids. Should be 3-p-Coumaroyalquinic acid.
Line 430. Should be “authentification”.
Author Response
Kind regards,
We thank you for your time and review and the suggestions you have expressed regarding the manuscript. All responses can be found in the attached file.
Kind regards

Reviewer 3 Report
Dear Authors,
The manuscript was modified according to reviewer’s suggestions and could be accepted to publication in Biology in the present form.
Yours sincerely,
Reviewer
Author Response
Kind regards,
We thank you for your time and review and the suggestions you have expressed regarding the manuscript.
Kind regards
Round 2
Reviewer 1 Report
Unfortunately, the authors did not fully address my concerns other than simply restating the text in many instances. They also did not revise the spelling errors in the one sentence that I indicated (line 122). Essenial should be essential and cosmetcs should be cosmetics. I will add further comments below.
Introduction
Lines 26 and 28: Should be phenolic contents or phenolics content
Line 28: Should be phenolic profiles
Line 30: essenial should be essential
Line 30: cosmetcs should be cosmetics
Line 30: Should farmacy be pharmacy? I’m just checking on these definitions.
Line 41 and 42: Should be phenolic contents (check throughout manuscript)
Line 43: Should be phenolic profiles (check throughout manuscript)
Line 49: Should this state “Walnuts are native”?
Line 50: Nowdays should be nowadays
Line 65: Should it state reuse?
Lines 76-77: Should the citations be separated?
Line 86: Should this be mold?
Lines 92-98: The authors indicated naphthoquinones have been identified in about 20 plant families. However, in their response document, they state that many naphthoquinones are unique to walnuts. These two sentences need to be clarified. Additionally, no references were provided in this paragraph. Which plant families contain naphthoquinones? Which new drugs can be developed? This brings up the point that this manuscript needs to emphasize the unique walnut polyphenols and the specific uses of these phenolics.
Line 99: Developed should be develop
Materials and Methods
Line 139: Therefore should be capitalized if it’s after a period
Line 195: Should be significance of the differences
Results and Discussion
Line 207: Should this be flavanols?
Lines 214-221: This paragraph seems to include a very significant finding from this study. This should be emphasized in the conclusion. Are these phenolics definitely unique to the walnut? Are there specific uses of these phenolics?
Line 384: There should be a period after the [19].
Conclusion
Line 419: This should be total phenolic contents
Line 428: This should be additives
The conclusion should state the unique walnut phenolics that were discovered and the specific uses for these phenolics (if applicable). This would add more significance to this study. I believe the authors might be missing an opportunity to emphasize the unique and important findings from this research study.
Author Response

(The authors gave the same response as above.)

Round 3
Reviewer 1 Report
I would like to thank the authors for addressing my concerns and emphasizing the significant findings from this research study.
Minor comments:
Line 50: I believe it should state, Walnuts are native
Line 213: I think flavanoles should be removed, as flavanols is stated on line 214